# LEARNING GEOMETRIC REASONING NETWORKS FOR ROBOT TASK AND MOTION PLANNING

**Smail Ait Bouhsain, Rachid Alami & Thierry Siméon**
Laboratory for Analysis and Architecture of Systems (LAAS)
National Center for Scientific Research (CNRS), Toulouse, France
`{saitbouhsa,alami,simeon}@laas.fr`

## ABSTRACT

Task and Motion Planning (TAMP) is a computationally challenging robotics problem due to the tight coupling of discrete symbolic planning and continuous geometric planning of robot motions. In particular, planning manipulation tasks in complex 3D environments leads to a large number of costly geometric planner queries to verify the feasibility of considered actions and plan their motions. To address this issue, we propose Geometric Reasoning Networks (GRN), a graph neural network (GNN)-based model for action and grasp feasibility prediction, designed to significantly reduce the dependency on the geometric planner. Moreover, we introduce two key interpretability mechanisms: inverse kinematics (IK) feasibility prediction and grasp obstruction (GO) estimation. These modules not only improve feasibility predictions accuracy, but also explain why certain actions or grasps are infeasible, thus allowing a more efficient search for a feasible solution. Through extensive experimental results, we show that our model outperforms state-of-the-art methods, while maintaining generalizability to more complex environments, diverse object shapes, multi-robot settings, and real-world robots.

## 1 INTRODUCTION

Task and Motion Planning (TAMP) (Garrett et al., 2021) is a robotics problem in which the goal is to find a sequence of robot actions and their corresponding motions to transition an environment from an initial state to a goal state. In most cases, the order of actions, their grounded parameters, and the feasibility of the corresponding motions are tightly coupled, requiring a careful combination of symbolic task planning and continuous geometric planning. However, the resulting large search space leads to a combinatorial explosion. Moreover, every action considered must be validated geometrically to ensure feasibility, leading to numerous costly queries to the geometric planner. Thus, geometric planning can become a bottleneck for TAMP (Bouhsain et al., 2023b).

Previous works (Wells et al., 2019; Driess et al., 2020a; Khodeir et al., 2023a) demonstrate that learning methods can help accelerate TAMP by providing fast geometric feedback to the task planner. In particular, action feasibility prediction offers an efficient alternative to geometric planning during the TAMP process. In offline manipulation planning, it can answer critical questions such as which object can be picked, how to grasp it, where to place it, or how to free access to it. However, action feasibility prediction presents several challenges. A suitable representation of 3D environments is needed, as these may contain an arbitrary number of objects, along with action representations that capture varying parameters such as grasps. Furthermore, predictions must not only be fast and accurate but also interpretable to understand why an action is infeasible and how to rectify it (e.g. another object is blocking access to the desired one). Finally, action feasibility prediction must generalize to environments with numerous objects, objects of varying shapes, and multi-robot settings.

Existing approaches to action feasibility prediction often struggle with interpretability, scalability, and generalization across diverse environments. To address these limitations, we propose a novel approach that leverages a GNN-based model for robot action and grasp feasibility prediction. Our method constructs a graph representation of 3D environments, where fixed and movable objects are represented as nodes, and edges capture spatial relationships and interaction constraints. Through this graph-based structure, we leverage an Edge-Enhanced Graph Attention Network (EGAT) to

predict action and grasp feasibility for each movable object. A unique aspect of our approach is the introduction of two interpretability mechanisms: inverse kinematics (IK) feasibility predictions, which determine whether the robot can feasibly manipulate an object from different sides, and Grasp Obstruction (GO) predictions, which quantify how neighboring objects restrict access to grasps from different sides of the object. These interpretable features not only allow us to predict action infeasibility, they also explain why a specific action fails, enabling more efficient planning.

The contributions of this paper are threefold: (1) We propose a novel GNN-based model for efficient and accurate action and grasp feasibility prediction in complex 3D environments. (2) We introduce two mechanisms, inverse kinematics (IK) feasibility and grasp obstruction (GO) predictions, that improve feasibility prediction while enhancing its interpretability, hence task planning efficiency. (3) We provide comprehensive experiments showcasing our method's state-of-the-art (SOTA) performance, including evaluations of its interpretability and generalization capabilities. Our code is available at: `https://github.com/Smail8/geometric_reasoning_networks.git`

## 2 RELATED WORKS

### 2.1 TASK AND MOTION PLANNING

Task and Motion Planning (TAMP) combines discrete symbolic task planning with continuous geometric motion planning to achieve robotic manipulation goals. Early approaches (Alami et al., 1990; Koga & Latombe, 1994; Ahuactzin et al., 1998; Siméon et al., 2004) view the problem primarily from a geometric perspective. These approaches build manipulation graphs that map valid grasps, placements, and connecting motions. Although, they suffer from a combinatorial explosion as the number of objects increases. Later, multi-modal motion planning (Hauser & Latombe, 2010; Hauser & Ng-Thow-Hing, 2011) generalized these methods using constraint-based graphs, but the complexity of constructing these graphs for cluttered environments remains a challenge. Modern methods integrate symbolic task planning with geometric planners, using backtracking when actions are infeasible (Cambon et al., 2009; Srivastava et al., 2014; Lagriffoul et al., 2014; Dantam et al., 2016; Garrett et al., 2018). However, in complex environments, the large number of queries to the geometric planner causes significant computational overhead, leading to slow planning processes.

### 2.2 LEARNING FOR TASK AND MOTION PLANNING

Many recent works leverage learning methods to provide heuristics for TAMP. Chitnis et al. (2016) apply reinforcement learning (RL) to learn policies for task refinement. Xu et al. (2021) model both immediate and future affordances in a learned latent space. Some methods leverage GNNs to identify the smallest set of objects for solving a planning problem (Silver et al., 2021), learning inter-robot relations to verify subgoal satisfaction (Huang et al., 2023), or predict dynamic object interactions (Chen et al., 2023). Eisner et al. (2024) tackles the problem of precise relative object placement using an SE(3)-equivariant learning. Agia et al. (2023) learn task-agnostic policies for various robot skills and verify their feasibility. Other works learn to fully solve TAMP problems. Zhu et al. (2021) leverage a two-level scene graph for neuro-symbolic task planning and graph-based motion generation. McDonald & Hadfield-Menell (2022) propose an imitation learning method that mimics a TAMP solver. Lin et al. (2022) introduce a GNN-based policy architecture trained on expert demonstrations. Haramati et al. (2024) propose a goal-conditioned RL framework using an object-centric image representation of the 3D environment. These methods tend to be problem-dependent, needing further training on unseen manipulation problems. Wang et al. (2024), Lin et al. (2023); Huang et al. (2024b) leverage large language models (LLM) to perform TAMP, incorporating feedback from geometric planning failures or predicted geometric infeasibility into the LLM to iteratively refine the solution. Although, these methods require large resources for running LLMs.

### 2.3 ACTION AND GRASP FEASIBILITY PREDICTION FOR MANIPULATION PLANNING

Action and grasp feasibility prediction is a fast growing research topic, which aims at accelerating common TAMP algorithms by reducing their dependency on the geometric planner. Wells et al. (2019) pioneered this problem by proposing to use Support Vector Machines (SVM) as action and grasp feasibility classifiers. Though, this method is limited to simple environments with a fixed number of objects. To tackle this issue, Driess et al. (2020b;a); Xu et al. (2022) propose to use top-view

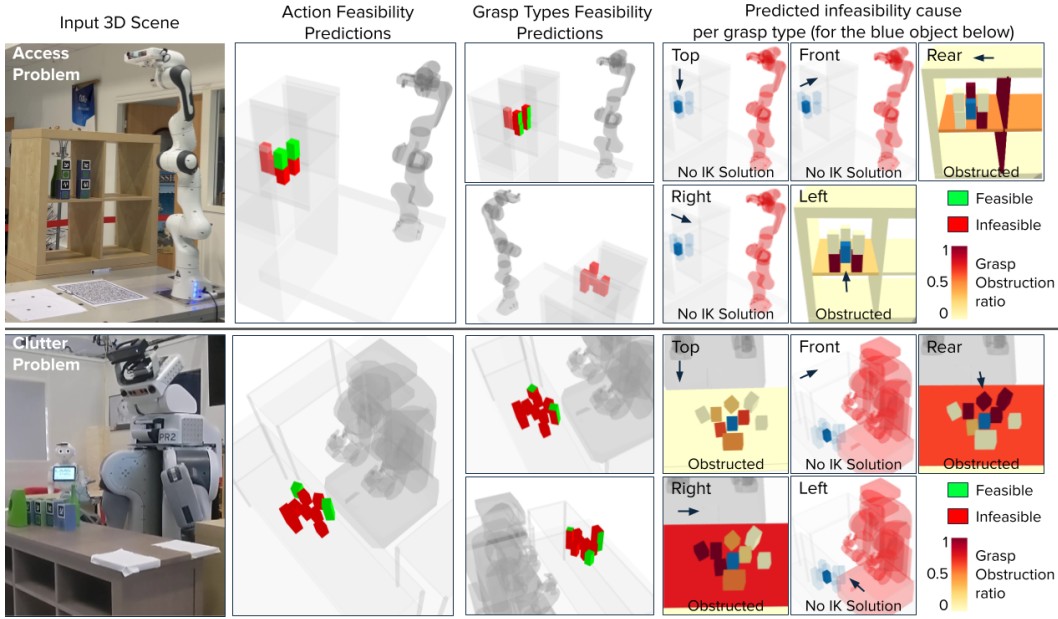

Figure 1: Visualization of **GRN** predictions on two manipulation problems, **Access** (Panda arm) and **Clutter** (PR2 Robot, predictions shown for its right arm). A single query to **GRN** outputs 3 predictions for each movable object in the environment: Action feasibility, grasp types feasibility (two views), and the predicted infeasibility causes for each grasp type. For clarity, we show the predicted infeasibility cause for one object only (shown in blue), and distinguish two cases: (1) *No IK solution*: robot shown in red, (2) *Grasp type obstructed*: we show all obstructing objects in a color gradient representing the obstructions ratio. Arrows show approach directions of grasp types.

depth images as input to a convolutional neural network (CNN). However, this approach is limited to tabletop problems. Bouhsain et al. (2023a;b) generalize this method to 3D environments by using 5 depth images from different scene views. Bouhsain et al. (2024) extend this method to mesh objects and multi-robot problems. Yang et al. (2022) also use multiple views, combined with text descriptions of actions and predicates as input to a transformer. Such image-based representations suffer from occlusions, which can hurt feasibility prediction accuracy. Some methods use 3D voxel grids (Park et al., 2022) or pointclouds (Huang et al., 2024a) to represent 3D scenes. Though, they suffer from high inference times and are, hence, limited to environments with few objects. Other works such as Kim et al. (2022); Khodeir et al. (2023a;b); Sung et al. (2023) tackle this issue by representing 3D environments using scene graphs, leveraging GNNs to predict the success of geometric planning steps from prior search experiences. These methods, however, lack interpretability and can not provide feedback on why actions are infeasible.

# 3 PROBLEM DESCRIPTION

In this work, we address the problem of predicting action and grasp feasibility for offline manipulation planning in 3D environments. The goal is to determine whether a robot can successfully plan actions, precisely picking or placing objects, and which grasps allow their collision-free motions, while accounting for inverse kinematics constraints and potential grasp obstructions.

## 3.1 OFFLINE MANIPULATION PLANNING CONTEXT

Our approach is tailored for offline manipulation planning tasks in 3D environments, containing fixed objects and movable objects. We assume the shape, dimensions, and pose of all objects are fully known, and that all objects remain static unless moved by the robot. A pick action involves grasping a movable object, while a place action refers to placing it at a target pose. From a motion planning perspective, these actions are symmetrical (Wells et al., 2019; Bouhsain et al., 2023b).

Assuming they start and end at the same home configuration of the robot, a place action at a specific pose is the reversed pick action from that pose. Throughout this work, the term "action" refers to both pick and place actions. Planning an action first requires sampling grasps. For each grasp, an inverse kinematics (IK) solver computes the corresponding robot configurations. If one exists, it is checked for collisions before a motion planner computes the robot's full trajectory. This process involving grasp sampling, IK solving, collision checking, and motion planning is computationally expensive and incurs significant overhead during TAMP, especially in cluttered 3D environments.

We define $\mathcal{G} = \{Top, Front, Rear, Right, Left\}$ as a set of 5 grasp types, each one representing a subspace of grasps related to the side from which the object is grasped. We focus on axis-aligned grasps, such that each type is the continuous set of grasps for which the end-effector's axis is parallel to one of the object's principal axes in a specific direction. This representation is similar to those introduced in previous works (Wells et al., 2019; Driess et al., 2020b;a; Bouhsain et al., 2023a;b).

## 3.2 FEASIBILITY PREDICTION

This work aims to reduce the dependency on the complex geometric planning process involved in offline manipulation planning. Given an environment $E$ and an object of interest $\mathcal{O} \in E$, the goal is to predict both the feasibility $F_a(\mathcal{O}, E) \in \mathbb{R}$ of picking or placing $\mathcal{O}$ at its pose in $E$, and the feasibility of each grasp type, $\mathbf{F}_{\mathcal{G}} = [F_g(\mathcal{O}, E), \forall g \in \mathcal{G}] \in \mathbb{R}^5$.

Moreover, we aim to estimate the cause of infeasibility for each grasp type. We focus on two primary factors contributing to infeasibility: (1) the absence of a valid inverse kinematics (IK) solution for all grasps within a grasp type $\boldsymbol{\kappa}_{\mathcal{G}} = [\kappa_g(\mathcal{O}, E), \forall g \in \mathcal{G}] \in \mathbb{R}^5$, and (2) the obstruction of grasps by each neighboring objects, represented as the ratio $\boldsymbol{\rho}_{\mathcal{G}}$ of obstructed grasps:

$$\boldsymbol{\rho}_{\mathcal{G}} = [\rho_g(\mathcal{O}, \mathcal{O}'), \quad \forall g \in \mathcal{G}, \quad \forall \mathcal{O}' \in \mathcal{N}(\mathcal{O})] \quad \in \mathbb{R}^{5 \times |\mathcal{N}(O)|} \tag{1}$$

where $\mathcal{N}(\mathcal{O})$ denotes the distance-based neighborhood of $\mathcal{O}$ and $|\mathcal{N}(O)|$ its cardinality. In addition to helping explain infeasibility and providing insights into the constraints imposed by the environment, these predictions are also used in the feasibility prediction process. In summary, the task at hand is to learn two classification functions $f_F$, $f_\kappa$, and a regression function $f_\rho$ s.t.:

$$\begin{bmatrix} F_a \\ \boldsymbol{F}_{\mathcal{G}} \end{bmatrix} = f_F(\mathcal{O}, E, \boldsymbol{\kappa}_{\mathcal{G}}, \boldsymbol{\rho}_{\mathcal{G}}) \qquad \text{where} \qquad \boldsymbol{\kappa}_{\mathcal{G}} = f_\kappa(O, E) \quad \text{and} \quad \boldsymbol{\rho}_{\mathcal{G}} = f_\rho(O, E) \tag{2}$$

## 4 GEOMETRIC REASONING NETWORKS

We propose **Geometric Reasoning Networks (GRN)**, a three-module GNN-based neural network which takes as input a graph representation of the environment, and outputs the action and grasp types feasibility for each movable object, as well as inverse kinematics feasibility (IK) and grasp obstruction (GO) predictions cf. Figure 1. The main challenge of learning methods in a manipulation planning context is finding an appropriate representation of the 3D environment which can contain an arbitrary number of objects. Previously proposed image-based representations suffer from occlusions as the number of objects in the environment increases, which can significantly impact performance. In this paper, we tackle this issue by representing 3D scenes as graphs where nodes represent fixed and movable objects and edges represent geometric relationships between objects.

### 4.1 3D SCENE REPRESENTATION

Given an environment $E$, we construct a directed graph $(\mathcal{V}, \mathcal{E})$ where each node corresponds to an object in $E$, fixed or movable. Nodes have a feature vector $\mathbf{x} = [l, w, h, x, y, z, \theta]^T$ where $(l, w, h)$ are the length, width and height of the object's bounding box, $(x, y, z)$ represent the position of the object in the environment, and $\theta$ is the object's orientation w.r.t its $z$ axis. The position and orientation of the object are expressed in the reference frame of the base of the robot. This allows a straightforward generalization to multi-robot settings simply by switching the reference frame to the base of the robot of interest. For each node $u$ corresponding to a movable object $\mathcal{O}$, we add a self-loop edge $(u \rightarrow u) \in \mathcal{E}$, as well as a directed edge $(v \rightarrow u) \in \mathcal{E}$ from each node $v$ corresponding to a neighboring (fixed or movable) object $\mathcal{O}' \in \mathcal{N}(\mathcal{O})$. Two objects are considered neighbors if the euclidean distance between their closest points along the $(x, y)$ plane is lower than

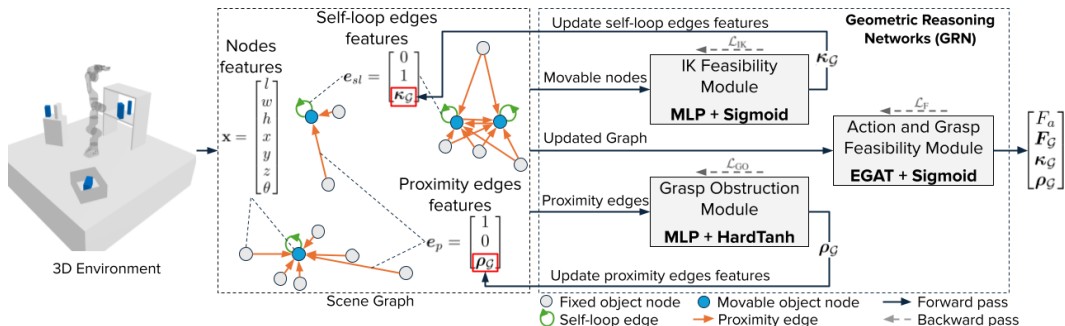

Figure 2: **Complete GRN architecture**. A scene graph is constructed from the input 3D environment. Node features of movable objects are given to IK feasibility module which outputs are used to update self-loop edge features. The concatenated features of nodes linked through a proximity edge are fed to the GO module to get grasp obstruction estimations, which are appended to proximity edge features. Finally, the updated graph is given to the AGF module to predict the action and grasp types feasibility for each movable object in the environment.

a user-defined threshold. Since our approach is GNN-based, where edges are used to determine which features to aggregate, the more edges are present in the input graph, the higher the number of computations is. Hence, the permissiveness of this threshold creates a tradeoff between accuracy and inference speed. Finally, self-loop edges and proximity edges are differentiated using a one-hot encoded feature vector such that $\boldsymbol{e}_{sl} = [0,1]^T$ and $\boldsymbol{e}_p = [1,0]^T$, where $\boldsymbol{e}_{sl}$ and $\boldsymbol{e}_p$ represent the feature vectors for a self-loop edge and a proximity edge respectively. The obtained graph is given as input to **GRN**, shown in Figure 2, which is comprised of three modules.

### 4.2 INVERSE KINEMATICS FEASIBILITY PREDICTION

The first module is the Inverse Kinematics (IK) feasibility prediction module. This submodel is a binary classifer, composed of a 4-layer Multi-Layer Perceptron (MLP) with ReLU activation functions, followed by a Sigmoid activation function. It takes as input the feature vector $\mathbf{x}_u$ of each node $u \in \mathcal{V}$ corresponding to a movable object, and simultaneously outputs the predicted inverse kinematics feasibility for each of the five grasp types in $\mathcal{G}$:

$$\boldsymbol{\kappa}_{\mathcal{G}}(u) = \text{Sigmoid}(\mathbf{MLP}_{\text{IK}}(\mathbf{x}_u)) \quad \in [0,1]^5 \tag{3}$$

Each prediction $\kappa_g$ where $g \in \mathcal{G}$ corresponds to the presence of a valid inverse kinematics solution for at least one grasp in $g$. This can be viewed as predicting whether the robot can reach a specific side of the object at its pose in the environment, without taking into account any other object. Since this information is valuable for action and grasp types feasibility prediction, we incorporate it to the previously constructed graph by concatenating the obtained predictions $\boldsymbol{\kappa}_{\mathcal{G}}$ to the features of self-loop edges resulting in $\boldsymbol{e}_{sl}^+ = [\boldsymbol{e}_{sl} \parallel \boldsymbol{\kappa}_{\mathcal{G}}] \in \mathbb{R}^7$, where $\parallel$ denotes the concatenation operator.

### 4.3 GRASP OBSTRUCTION ESTIMATION

The second stage is Grasp Obstruction (GO) estimation, which is a regression module consisting of 4-layer MLP with ReLu activation functions, followed by a HardTanh activation function which bounds the output between 0 and 1. It takes as input the concatenated features of each pair of nodes $(v \rightarrow u)$ linked through a proximity edge. For each grasp type $g \in \mathcal{G}$, it outputs the estimated ratio of grasps of $u$ that are obstructed by the object corresponding to $v$, which is defined as the number of obstructed grasps divided by the total number of grasps of a specific type:

$$\boldsymbol{\rho}_{\mathcal{G}}(u,v) = \text{HardTanh}(\mathbf{MLP}_{\text{GO}}([\boldsymbol{x}_u \parallel \boldsymbol{x}_v])) \quad \in [0,1]^5 \tag{4}$$

This submodel not only predicts whether an object obstructs grasps of another, it also gives insight into how much it blocks access to it. In a manipulation planning context, this allows to rank obstructing objects and plan accordingly. Similarly to IK feasibility predictions, this information impacts directly the feasibility of actions and grasp types. Thus, we concatenate the grasp obstruction estimations $\boldsymbol{\rho}_{\mathcal{G}}$ to the edge features of proximity edges to obtain $\boldsymbol{e}_p^+ = [\boldsymbol{e}_p \parallel \boldsymbol{\rho}_{\mathcal{G}}] \in \mathbb{R}^7$.

### 4.4 Action and Grasp Feasibility Prediction

Once our graph is constructed and enriched through IK feasibility and GO predictions, it is given to the action and grasp feasibility (AGF) prediction module. As mentioned previously, IK feasibility and grasp obstructions, which constitute our edge features, have a direct impact on action and grasp type feasibility. However, the classic graph attention network (GAT) (Veličković et al., 2018) uses edge features simply for computing attention weights, and does not take them into account during the aggregation process. This causes the model to lose valuable information when applied to tasks where edge features are as important as node features. In this work, we propose to use the Edge-Featured Graph Attention Network (EGAT) (Wang et al., 2021) which, in addition to using edge features to compute attention, leverages both node and edge features during the aggregation process.

The third submodel first computes embeddings $h_u$ and $e_{uv}^*$ of node and edge features using two fully-connected layers such that $h_u = W_h . x_u$ and $e_{uv}^* = W_e . e_{uv}^+$, where $W_h$, $W_e$ are weight matrices. Then, EGAT computes the multi-head attention of each edge as:

$$\alpha_{uv} = \frac{\exp(a^T \text{LeakyReLU}([h_u \parallel h_v \parallel e_{uv}^*]))}{\sum_{k \in \mathcal{N}(u) \cup \{u\}} \exp(a^T \text{LeakyReLU}([h_u \parallel h_k \parallel e_{uk}^*]))} \tag{5}$$

Note that this formulation is slightly different from the one proposed by Wang et al. (2021). We adapt the attention computation introduced by Brody et al. (2022), which fixes the static attention problem of the standard GAT. Once the multi-attention weights computed, our model computes a weighted average of the concatenated node and edge embeddings, followed by a LeakyReLU activation:

$$h_u' = \text{LeakyReLU}\left(\sum_{v \in \mathcal{N}(u) \cup \{u\}} \alpha_{uv}[h_v \parallel e_{uv}^*]\right) \tag{6}$$

Finally, the obtained vector is passed through a 2-layer MLP followed by a sigmoid activation function that outputs the action and grasp types feasibility predictions for movable object node $u$:

$$\begin{bmatrix} F_a(u) \\ \mathbf{F}_\mathcal{G}(u) \end{bmatrix} = \text{Sigmoid}(\mathbf{MLP}_F(h_u')) \tag{7}$$

### 4.5 Training Strategy

Our proposed **GRN** model is trained in a supervised manner by first pre-training each module separately. The IK feasibility prediction submodel is trained using a binary cross entropy loss denoted $\mathcal{L}_{\text{IK}}$ while the GO estimation module is trained using the mean square error loss $\mathcal{L}_{\text{GO}}$. Regarding the AGF prediction module, it is trained using a binary cross entropy loss $\mathcal{L}_{\text{F}}$, using the ground truth of IK feasibility and grasp obstructions as edge features. The complete **GRN** model is then fine-tuned using a weighted sum of the previously defined losses such that:

$$\mathcal{L} = \mathcal{L}_{\text{F}} + \mathcal{L}_{\text{IK}} + \eta . \mathcal{L}_{\text{GO}} \tag{8}$$

where $\eta$ is a weighting factor allowing a balanced order of magnitude across classification and regression. This training strategy is inspired by the one proposed by Chen et al. (2020).

During training, our 3D scene representation allows two data augmentation methods: dimensions switch and rotation. They both take advantage of the symmetry of bounding boxes. Indeed, switching the length and width of a bounding box, then applying a $\frac{\pi}{2}$ rotation around the $z$ axis, keeps the geometric properties of the environment unchanged, resulting in a new node feature vector $\boldsymbol{x} = [w, l, h, x, y, z, \theta \pm \frac{\pi}{2}]^T$. The same goes for applying a $\pi$ rotation without any dimensions switch such that $\boldsymbol{x} = [l, w, h, x, y, z, \theta \pm \pi]^T$. These can be applied to fixed and movable objects. For the latter case, the labels associated with grasp types need to be interchanged, since switching the dimensions and/or rotating the object changes which side is considered as the front for example.

## 5 Experiments

We conduct a series of experiments in order to evaluate the performance of our proposed method compared to existing approaches, and showcase the generalization capabilities of our approach. We

conduct our experiments using mainly the Franka Emika Panda, which is a 7 degrees-of-freedom (DOF) robotic arm with a parallel jaw gripper. Since our model is robot-centric, meaning that it is specific to the robot it is trained for, we showcase the applicability of our method to other robotic manipulators by running experiments on the Willow Garage PR2 robot as well. The latter is a dual-arm robot where each arm has 7 DOFs and a parallel jaw gripper, and a telescoping spine.

## 5.1 DATASETS

Our model is trained and evaluated on fully synthetic data. We generate a number of datasets following the method described in Appendix B. Each one characterized by a number of movable objects as well as a minimum and maximum numbers of structures (e.g, rack, counter, basket) and obstacles. They are also characterized by the robot used during data annotation.

**Panda-3D-4:** This is dataset is composed of 3D environments containing 4 movable objects, 1 to 4 structures and 0 to 4 obstacles and is annotated using a Panda robot.

**Panda-Tabletop-4**: In order to conduct a fair comparison to tabletop methods Wells et al. (2019); Driess et al. (2020b), we generate a dataset consisting of tabletop environments with 0 structures, 4 movable objects and up to 4 obstacles, all placed on the same support surface as the robot's base.

**PR2-3D-4**: We generate this dataset to showcase the applicability of our approach to other robotic arms as well as multi-robot settings. It is generated using the same parameters as the **Panda-3D** dataset, and annotated for the right arm of PR2 robot[1]. We consider that the base of robot is fixed to the ground, and that the telescopic spine is one of the DOFs of the arms.

In order to quantitatively measure the generalizability of our approach to environments containing a higher number of objects than training environments, we generate three additional test sets denoted **Panda-3D-10**, **Panda-3D-15**, **Panda-3D-20**, each composed of 1'000 environments containing 10, 15 and 20 movable objects respectively. These environments also contain a higher number of fixed objects, with a number of structures ranging from 4 to 8, as well as 2 to 4 obstacles. Moreover, we increase the dimensions' range used during data generation resulting in larger objects.

## 5.2 BASELINES

We compare our proposed approach to multiple baselines and methods proposed or adapted from previous works on action and grasp feasibility prediction for manipulation planning.

**MLP:** This is a simple baseline which uses a 4-layer MLP that takes as input the feature vector $x$ of an object to predict action and grasp feasibility, without considering the rest of the environment.

**Feasibility-SVM (F-SVM):** Introduced by Wells et al. (2019), this method uses multiple SVMs to predict the action and grasp types feasibility prediction in tabletop environments containing 2 movable objects represented using hand-crafted feature vectors.

**Deep Visual Heuristics (DVH):** This method, proposed by Driess et al. (2020b), represents environments using top-view depth images, then uses a CNN to predict the feasibility of an action using a grasp type. We adapt **DVH** to output the feasibility of the action and all grasp types simultaneously.

**Action and Grasp Feasibility Prediction Network (AGFP-Net):** Bouhsain et al. (2023b) propose this method as an extension of **DVH** to 3D environments, using 5 depth images corresponding to different views of the scene as input to the CNN.

**Feasibility-GAT (F-GAT):** This baseline is an adapted version of the methods proposed by Silver et al. (2021) and Khodeir et al. (2023a;b). It represents environments as graphs where nodes represent objects, and edges represent symbolic relationships (e.g object on table). Nodes features are the dimensions and pose of objects and edge features are one-hot-encodings of the different relationship types. GAT is then used to predict action and grasp feasibility.

**Feasibility-GCN (F-GCN):** This baseline uses the same scene representation as **F-GAT**, except that GAT is replaced with a Graph Convolution Network (GCN), which does not use edge features.

---

[1]As explained is Section 4.1, our model can be applied to different arms of the same type simply by expressing the objects' poses in the frame of reference of the considered arm.

Table 1: Comparison with SOTA methods trained and tested on different datasets. For grasp types feasibility prediction, the mean ($\pm$ standard deviation) of F1 scores of the 5 grasp types are reported.

| Dataset | Panda-3D-4 | | Panda-Tabletop-4 | | PR2-3D-4 | |
|---|---|---|---|---|---|---|
| Task | Action (F1) | Grasp (F1) | Action (F1) | Grasp (F1) | Action (F1) | Grasp (F1) |
| F-SVM | - | - | 0.884 | 0.415 ($\pm$ 0.220) | - | - |
| MLP | 0.784 | 0.558 ($\pm$ 0.089) | 0.911 | 0.696 ($\pm$ 0.121) | 0.750 | 0.574 ($\pm$ 0.104) |
| DVH | 0.840 | 0.718 ($\pm$ 0.108) | 0.961 | 0.865 ($\pm$ 0.073) | 0.808 | 0.622 ($\pm$ 0.179) |
| AGFPNet | 0.882 | 0.806 ($\pm$ 0.065) | 0.964 | 0.916 ($\pm$ 0.032) | 0.836 | 0.655 ($\pm$ 0.238) |
| F-GCN | 0.836 | 0.721 ($\pm$ 0.057) | 0.955 | 0.879 ($\pm$ 0.042) | 0.791 | 0.680 ($\pm$ 0.075) |
| F-GAT | 0.867 | 0.796 ($\pm$ 0.052) | 0.961 | 0.904 ($\pm$ 0.034) | 0.827 | 0.764 ($\pm$ 0.061) |
| GRN (Ours) | **0.939** | **0.940** ($\pm$ **0.009**) | **0.976** | **0.976** ($\pm$ **0.004**) | **0.908** | **0.903** ($\pm$ **0.013**) |

## 5.3 Evaluation Metrics

We use the **F1** score as the evaluation metric for classification tasks, namely action, grasp types and IK feasibility predictions. For or Grasp Obstruction (GO) predictions, we use the **Mean Absolute Error** to measure performance. For clarity, we report the mean and standard deviation of predictions across the different grasp types.

## 6 Results

### 6.1 Comparison to Prior Work

Table 1 shows that our proposed model outperforms all prior works on both action feasibility and grasp types feasibility predictions, and on all datasets. CNN-based methods, **DVH** and **AGFP-Net**, fall short compared to our approach, with a difference in F1 score on the **Panda-3D-4** of $10\%$ (resp. $5.7\%$) for action feasibility prediction, and $22.6\%$ (resp. $13.7\%$), for grasp type feasibility prediction. Indeed, image-based scene representation suffers from occlusions due to the 3D nature of the environment, resulting in inaccurate predictions for occluded objects. GNN-based methods, on the other hand, represent 3D environments using scene graphs. However, the performance of our approach compared to **F-GCN** and **F-GAT** shows that a careful design of the graph and its connectivity is needed. Silver et al. (2021) and Khodeir et al. (2023a;b) connect nodes using symbolic facts. Our method uses geometric relationships between objects to connect the graph. This allows our model to achieve an F1 score up to $10.3\%$ higher than other GNN-based baselines on action feasibility prediction, and up to $21.8\%$ higher on grasp types feasibility prediction on **Panda-3D-4**.

Comparing the standard deviations across F1 scores of each grasp type shows that our proposed method has a more consistent performance across the different grasp types than other models. This is due to our data augmentation method and the two interpretation mechanisms. Switching the dimensions then/or rotating an object implies interchanging grasp types annotations, which ensures balanced labels across grasp types. This method is not applicable to CNN-based methods [2]. Additionally, the use of IK feasibility predictions and GO estimations as edge features allows each grasp type feasibility prediction to be informed, which is not the case for other GNN-based methods.

In robotic manipulation planning, feasibility prediction must not only be accurate, it must also have a low inference time and memory footprint. Table 2 reports the number of parameters in previous models compared to **GRN**, as well as the inference time of each model on a 3D scene containing a Panda robot, 4 movable objects and 15 fixed object. The inference time incorporates the complete prediction process from the model's input construction to the output, for each movable object in the environment. For reference, we also report the planning time of an off-the-shelf geometric planner.

Table 2: Comparison of the number of parameters and inference time on a 3D environment with 4 movable objects and 15 fixed objects (4 queries).

| Model | Inference time (ms) | Nb Parameters |
|---|---|---|
| Geometric planner | 1500 | - |
| MLP | 0.6 | 269'830 |
| DVH | 25 | 11'225'282 |
| AGFPNet | 150 | 34'585'350 |
| F-GCN | 4.25 | 1'057'798 |
| F-GAT | 5.0 | 2'636'806 |
| GRN (Ours) | 5.5 | 2'259'472 |

---

[2] Switching the dimensions then rotating an object results in the same input images.

Table 3: Ablation Study on the **Panda-3D-4** dataset. For each task related to grasp types, we report the mean ($\pm$ standard deviation) across all grasp types.

| Task | Action (F1) $\uparrow$ | Grasp (F1) $\uparrow$ | IK (F1) $\uparrow$ | GO (MAE) $\downarrow$ |
|---|---|---|---|---|
| w/o IK, w/o GO | 0.868 | 0.811 ($\pm$ 0.043) | - | - |
| w/o GO | 0.872 | 0.811 ($\pm$ 0.046) | **0.995 ($\pm$ 0.001)** | - |
| w/o IK | 0.937 | 0.933 ($\pm$ 0.011) | - | 0.029 ($\pm$ 0.003) |
| Full model w/ GAT | 0.928 | 0.924 ($\pm$ 0.013) | **0.995 ($\pm$ 0.001)** | 0.029 ($\pm$ 0.003) |
| Full model w/o data aug. | 0.915 | 0.903 ($\pm$ 0.013) | 0.990 ($\pm$ 0.001) | 0.044 ($\pm$ 0.002) |
| Full model (Trained from scratch) | 0.932 | 0.925 ($\pm$ 0.011) | 0.994 ($\pm$ 0.001) | 0.038 ($\pm$ 0.002) |
| **Full model (Ours)** | **0.939** | **0.939 ($\pm$ 0.009)** | **0.995 ($\pm$ 0.001)** | **0.028 ($\pm$ 0.003)** |

Results show that, compared to previous works, our method yields the most accurate predictions while being one of the models with the lowest inference times and memory footprints. Furthermore, **GRN** has a 99.6% lower inference time than traditional geometric planning. In an offline manipulation planning context, where the number of feasibility checks can reach tens of thousands of queries, this difference in computational cost can significantly reduce planning time.

## 6.2 ABLATION STUDY

In order to justify the choices behind our neural network architecture and training strategy, we conduct an ablation study on the **Panda-3D-4** dataset. Results reported in Table 3 showcase the importance of the proposed interpretation modules. Our full model shows a 7.1% gain in performance compared to the one without IK feasibility and GO predictions. A more in-depth analysis shows that the grasp obstruction estimation module is the most important, while IK feasibility prediction yields a slight improvement in performance. Moreover, the improved performance obtained using EGAT instead of classical GAT Veličković et al. (2018) shows that incorporating edge features in the attention computation and the aggregation process helps action and grasp types feasibility prediction. We conduct two ablations to demonstrate the effectiveness of our training strategy. Training the model without the proposed data augmentation method, yields a lower performance on all tasks, particularly on grasp types feasibility prediction and GO estimation with a 3.6% difference in F1 score, and 1.6 % in MAE. Finally, training the full model from scratch, rather than pre-training each module before fine-tuning the complete network, yields a slightly lower performance across all tasks.

## 6.3 GENERALIZABILITY EVALUATION

**Applicability to other robots.** Table 1 shows our model's performance on the **PR2-3D-4** compared to prior methods. The results show that **GRN** achieves a better performance than the state-of-the-art on robots with various kinematics. A consistent decrease in performance can be noticed across all methods (ours included), compared to when trained on the **Panda-3D-4** dataset. This is due to the smaller number of training data of the PR2 dataset and the harder kinematics of the PR2 robot.

**Generalizability to more complex environments.** Table 4 reports the performance of our method and the baselines on the **Panda-3D-10**, **Panda-3D-15** and **Panda-3D-20** test sets, when trained on the **Panda-3D-4** dataset. Results show that, although there is a decrease in F1-scores compared to the one obtained on **Panda-3D-4**, **GRN** maintains a good performance on 3D environments with a higher number of fixed and movable objects, with an F1-score on **Panda-3D-20** of 0.89 for action feasibility prediction, and 0.903 for grasp types feasibility prediction. Particularly, our model shows a better generalization capability on the latter than previous CNN-based or GNN-based methods.

## 6.4 APPLICATION TO TASK AND MOTION PLANNING

Although the integration of our method into a task and motion planner is outside the scope of this paper, we develop a simple single-shot planner (cf. Appendix C.1) to showcase the power of **GRN**'s predictions and its interpretation mechanisms, as well as the planning performance gain yielded by our approach. From a single query to **GRN**, it proposes a geometrically feasible plan to a class of manipulation planning problems, where the goal is to move a single object initially placed in a complex setting. We compare this algorithm to the non-informed TAMP planner proposed by

Table 4: Evaluation of the generalizability to 3D environments with a higher number of objects compared to SOTA methods, when trained on the **Panda-3D-4** dataset.

| Test set | Panda-3D-10 | | Panda-3D-15 | | Panda-3D-20 | |
|---|---|---|---|---|---|---|
| Task | Action (F1) | Grasp (F1) | Action (F1) | Grasp (F1) | Action (F1) | Grasp (F1) |
| MLP | 0.773 | 0.624 ($\pm$ 0.028) | 0.766 | 0.616 ($\pm$ 0.046) | 0.768 | 0.609 ($\pm$ 0.047) |
| DVH | 0.820 | 0.697 ($\pm$ 0.115) | 0.819 | 0.686 ($\pm$ 0.127) | 0.825 | 0.676 ($\pm$ 0.140) |
| AGFPNet | 0.858 | 0.770 ($\pm$ 0.079) | 0.862 | 0.768 ($\pm$ 0.078) | 0.864 | 0.755 ($\pm$ 0.087) |
| F-GCN | 0.794 | 0.633 ($\pm$ 0.036) | 0.771 | 0.595 ($\pm$ 0.044) | 0.764 | 0.565 ($\pm$ 0.055) |
| F-GAT | 0.829 | 0.738 ($\pm$ 0.026) | 0.826 | 0.725 ($\pm$ 0.032) | 0.825 | 0.715 ($\pm$ 0.038) |
| **GRN (Ours)** | **0.894** | **0.909 ($\pm$ 0.012)** | **0.891** | **0.906 ($\pm$ 0.013)** | **0.890** | **0.903 ($\pm$ 0.012)** |

Table 5: Performance of **GRN** planner compared to a non-informed planner on the **Access** and **Clutter problems**. Results are average over 10 runs on 10 different instances of each problem.

| Problem | Method | Success Rate (%) | Planning time (s) | Nb Geometric Planner Calls |
|---|---|---|---|---|
| Access | Bouhsain et al. (2024) | 100% | 26.5 | 41.1 |
| | **GRN planner** | 90% | 3.17 | 6 |
| Clutter | Bouhsain et al. (2024) | 100% | 558.9 | 89.5 |
| | **GRN planner** | 100% | 14.8 | 7.2 |

Bouhsain et al. (2024) on two different problems shown in Figure 1. The first is the **Access** problem, where a single Panda robot has to move a small bottle, to which a number of fixed and movable objects block access. The second is a multi-robot **Clutter** problem, in which both arms of the PR2 robot can collaborate to pick an object surrounded by grasp-obstructing objects. Objects used in these problems are a mix of box-shaped and mesh objects. Table 5 shows that despite its simplicity, the **GRN**-based planner not only achieves a 90% and 100% success rate but also reduces planning time by 88% on the **Access** problem, and 97% on the **Clutter** problem. This is possible thanks to the two interpretation mechanisms, which allow the planner to reason over why an action or a grasp type is infeasible, and recursively decide which objects should be moved to rectify that. We also test **GRN** planner in real-world setups, on both a Panda and a PR2 robots. Objects' poses are estimated using a separate perception module. The proposed model is able to accurately predict action and grasp feasibility, as well as reasons of infeasibility, from estimated objects' poses in both setups, allowing the planner to compute feasible solutions that are successfully executed by the robots.

## 7 DISCUSSION AND FUTURE WORK

In this work, we propose a framework for action and grasp feasibility prediction in 3D environments. Leveraging a GNN-based neural network and two interpretation mechanisms, our model predicts the feasibility of pick or place actions, different grasp types, and the infeasibility cause for each grasp type from a scene graph representation. Results demonstrate that our approach outperforms state-of-the-art methods, generalizing better to complex environments and robots. Additionally, through IK feasibility and grasp obstruction predictions, a class of manipulation planning problems can be solved with a single query to our model, significantly reducing planning time compared to traditional TAMP planners. Our design also enables straightforward extensions to diverse object shapes via bounding boxes and multi-robot feasibility predictions due to the robot-centric nature of **GRN**.

Currently, our method is limited to infeasibility caused by missing IK solutions or grasp obstructions, and it struggles with problems stemming from infeasible robot motion. Future work will include graph pooling layers to evaluate motion infeasibility across the entire scene graph. Additionally, grasp obstructions are currently estimated as a blocked ratio, which does not indicate which grasps are obstructed. Representing obstructions as blocked object regions could improve interpretability. While bounding boxes work well for common objects, they face challenges with large or complex shapes. A potential extension is using multi-bounding box representations, treating sub-boxes as separate nodes. Alternatively, embeddings from off-the-shelf shape encoders (e.g., meshes or point clouds) could enhance node features, though computational efficiency remains a challenge. Finally, we also aim at integrating our model into a more sophisticated TAMP algorithm.

## ACKNOWLEDGMENTS

This work is partially supported by the EU-funded project euROBIN under grant agreement no. 101070596 (https://www.eurobin-project.eu/). We also would like to thank Baptiste Merlau for his help with the real-world experiments.

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

## A  IMPLEMENTATION DETAILS

**Scene Graph Construction**. Scene graphs are built using the graph representation from Pytorch Geometric (Fey & Lenssen, 2019). Nodes correspond to fixed and movable objects from the environment. Self-edge loops are added for nodes corresponding to movable objects. When adding proximity edges, since evaluating the distance between two objects closest points can be computationally expensive, we consider two objects $\mathcal{O}$ and $\mathcal{O}'$ as neighbors if the euclidean distance between their centers is lower than a threshold $\epsilon = r_\mathcal{O} + r_{\mathcal{O}'} + K$, where $r_\mathcal{O}$ and $r_{\mathcal{O}'}$ are the bounding cylinder radii of $\mathcal{O}$ and $\mathcal{O}'$ respectively, and K is constant set to 0.6. This constant is chosen as twice the length of the last link of the robot. Using this definition, the neighborhood of an object is easy to compute and captures enough of its surroundings for accurate action/grasp feasibility prediction.

**Model Architecture.** The IK feasibility prediction and GO estimation modules are 4-Layer MLPs with a hidden size of 512. The AGF module, on the other hand, has a hidden size of 256, 4 attention heads and one message-passing step. Exceptionally, when training on the **PR2-3D-4** dataset, we use a hidden size of 256 for the GO module as it yields better results.

**Training Details.** The three modules are implemented in Pytorch Geometric (Fey & Lenssen, 2019) and trained using the Adam optimizer (Kingma, 2014). During the pre-training stage, each module is trained for 100 epochs. We use a batch size of 8192 and a learning rate of 0.001 for the IK feasibility classifier and the GO estimator. For AGF classifier, we set the batch size to 2048 and the learning rate to 0.0001. During the fine-tuning stage, the complete **GRN** model is trained for 100 epochs with a batch size of 2048 and a learning rate of 0.0001. The model is trained on an Intel(R) Xeon(R) W-2223 CPU @ 3.60GHz workstation, with an NVIDIA RTX A5000 GPU. The full training process takes approximately 15 hours.

**Inference Time Decomposition.** Table 6 shows the inference time decomposition of our proposed model. The total inference time of **GRN** is 5.5 ms in average, with the most significant portion spent on scene graph construction with an average time cost of 3 ms. During this step, time is mostly spent on evaluating the distance between pairs of objects to add proximity edges between their corresponding nodes. The IK feasibility and GO modules take each a computation time of 0.5 ms. Finally, the AGF module takes 1.5 ms to output the action and grasp feasibility prediction modules.

Table 6: Decomposition of the inference time of **GRN** on a 3D environment with 4 movable objects and 15 fixed objects.

| Step | Inference Time |
|---|---|
| Scene Graph Construction | 3 ms |
| IK Feasibility Module | 0.5 ms |
| GO Module | 0.5 ms |
| AGF Module | 1.5 ms |
| Total | 5.5 ms |

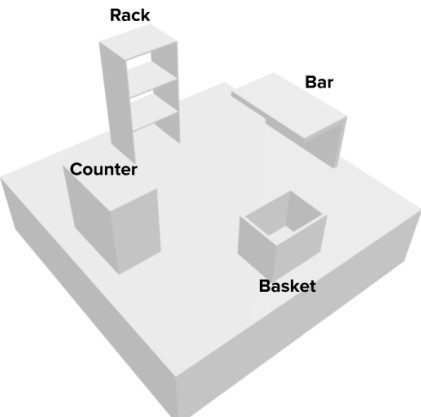

Figure 3: Visualization of the different types of structures used during data generation.

## B  DATA GENERATION AND ANNOTATION METHOD

**Data Generation.** Our data generation process consists of generating a number of 3D environments containing a random number of support surfaces, movable objects and obstacles within a specified range. We define 4 types of structures as shown in Figure 3: (1) a rack which is a structure containing a varying number of shelves separated by a gap, and a holder on each side or each corner, (2) a bar which is a L-shaped structure with one support surface and one holder, (3) a basket which consists of a support surface and 4 sides surrounding it, and (4) a counter which is a large block. When generating an environment, we first randomly sample a number of structures within the specified range. We then choose one of the predefined structures, before randomly sampling its dimensions and pose. After collision checking using the FCL library (Pan et al., 2012), all objects composing the structure are added to the environment as fixed objects. These can be either support surfaces or obstacles (e.g holders). This process is repeated until the number of structures sampled is reached.

Given the fact that our approach represents objects using their bounding boxes, it is sufficient to generate a dataset consisting of box-shaped objects only. After sampling a number of movable objects to add, we randomly sample the dimensions of the object with a specified range. In order to incorporate difficulties encountered in real-world scenarios, we define three methods for object placement sampling: (1) the first is random placement in which a support surface and a pose within its bounds are randomly sampled, (2) proximity placement in which the object is placed in the neighborhood of a randomly chosen object, (3) the third is the underneath placement in which the object is placed underneath a randomly chosen support surface. We check the sampled object placement for collisions before adding it to the environment. We then repeat the process until the sampled number of movable objects is reached. Finally, a number of fixed box-shaped obstacles is randomly added to the scene, before storing it.

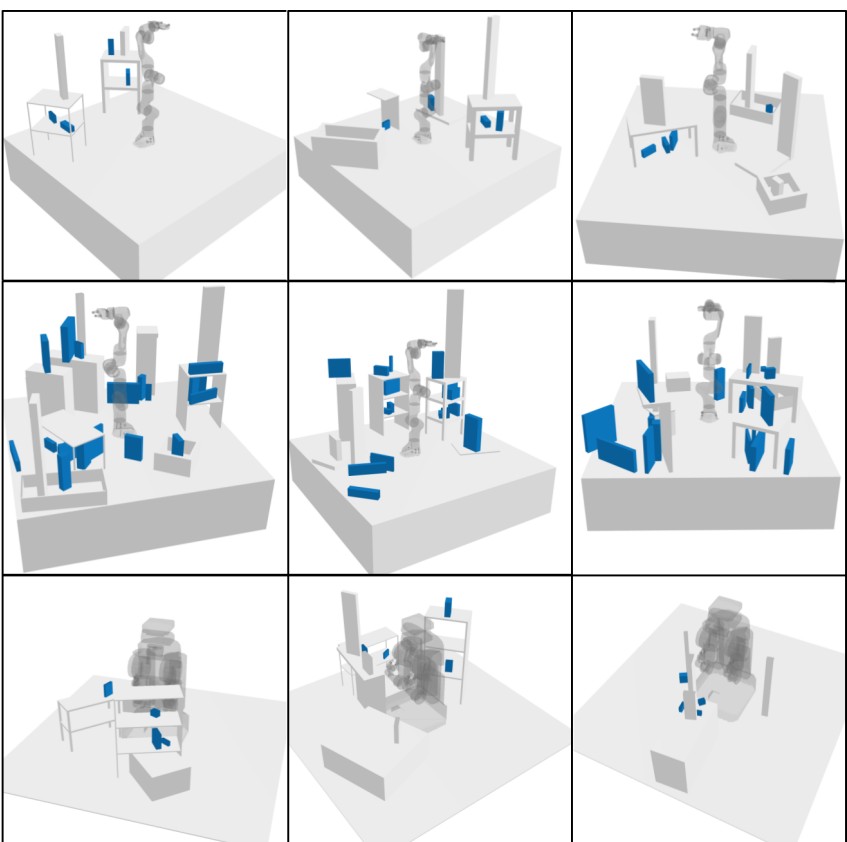

Figure 4: Visualization of generated environments from the (top) **Panda-3D-4**, (middle) **Panda-3D-20**, and (bottom) **PR2-3D-4** test sets

Using this method, we generate the datasets presented in Section 5.1. The **Panda-3D-4** dataset consists of a training set containing 70'000 scenes, a validation set of 10'000 scenes, and a test set of 20'000 scenes, each one generated using a different random seed to ensure that the environments are different across all three sets. The **Panda-Tabletop-4** and **PR2-3D-4** are each composed of 25'000 training scene, 5'000 validation scenes and 10'000 test scenes. Figure 4 shows environments from different datasets.

**Data Annotation.** Data annotation can be done using any off-the-shelf geometric planner. In this work, we use an adapted version of Moveit Task Constructor (Görner et al., 2019), with the KDL plugin for IK computation, FCL (Pan et al., 2012) for collision checking and, Bidirectional Transition-Based Rapid Random Tree (Devaurs et al., 2013) for motion planning. Since pick and place actions are symmetrical, they can be considered as equivalent. Thus, we only annotate pick actions. Also, we do not focus on trivial infeasibility cases due to the object being larger than the gripper. Hence, during annotation, we allow collisions between the object to pick and the gripper fingers. This also allows the handling of mesh objects such as a mug or a wine glass, for which feasible grasps exist even if the bounding box is larger than the gripper's maximum width.

For each movable object in a generated environment, we query the geometric planner to plan a pick action from its placement in the environment. First, we uniformly sample a number of grasps depending on the size of the object. For each sampled grasp, we compute up to 8 IK solution, which are then checked for collisions. Finally, the motion planner is queried for each collision-free IK solution until a feasible collision-free trajectory is found. If a solution is found, the action is annotated as feasible. In parallel, we set the feasibility label of each grasp type $g \in \mathcal{G}$ to 1 if at least one grasp in $g$ is feasible. Similarly, we set the IK feasibility label of a grasp type to 1 if at least one grasp belonging to it has a valid inverse kinematics solution. Finally, we record all grasp-obstructing objects as well as the ratio of grasps obstructed per grasp type. Data annotation takes approximately 68 hours to complete on the **Panda-3D-4** dataset.

Figures 5 and 6 show the distributions of obtained labels for the **Panda-3D-4** and **PR2-3D-4** training sets. For the former, the action feasibility annotations are balanced, while grasp types feasibility is imbalanced with more infeasible cases. Regarding reasons of infeasibility, Figure 5c shows a balanced distribution between failures due to IK feasibility and grasp obstructions. Cases where

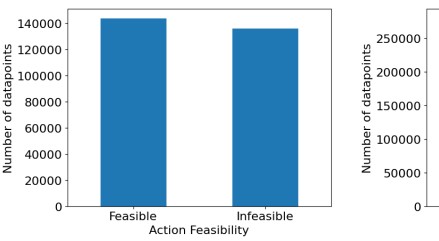
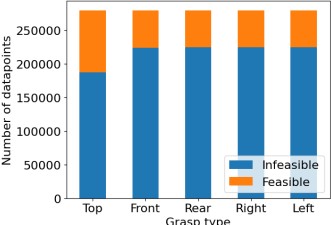
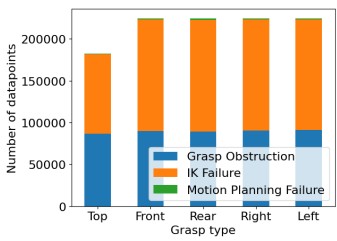

(a) Number of feasible and infeasible actions

(b) Number of feasible and infeasible cases per grasp type

(c) Distribution of failure causes per grasp type

Figure 5: Annotations statistics for the **Panda-3D-4** training set.

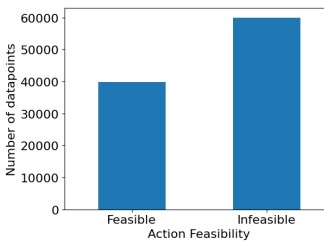
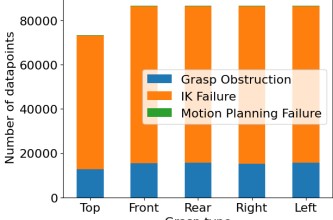

(a) Number of feasible and infeasible actions

(b) Number of feasible and infeasible cases per grasp type

(c) Distribution of failure causes per grasp type

Figure 6: Annotations statistics for the **PR2-3D-4** training set.

infeasibility is due to motion planning failure, on the other hand, appear rarely in the dataset. The **PR2-3D-4** dataset has a more pronounced imbalance towards infeasibility, the latter being caused more often by the absence of inverse kinematics solutions than grasp obstructions, showcasing the more complex kinematics of the PR2 robot.

## C APPLICATION TO TAMP

### C.1 GRN PLANNING ALGORITHM

Algorithm 1 shows the pseudo-code for the GRN-based planner. Given the **GRN** predictions on the initial state of the environment, a feasibility threshold and a grasp obstructions threshold, it recursively builds a task plan by moving objects when it is feasible, or trying to free access to grasp types based on grasp obstruction and IK feasibility information otherwise. Once a task plan is found, we query the geometric planner used during data annotation to verify its geometric feasibility, and plan the corresponding robot motions. For simplicity, Algorithm 1 shows the planner used for the single robot problem only. For the multi-robot **Clutter** problem using both arms of the PR2 robot, we choose the robot with the higher feasibility prediction to execute feasible actions. If an action is infeasible for both arms, we consider both of them when trying to free grasp types.

---

**Algorithm 1** GRNPlanner

---

**Input:** $F_a, \boldsymbol{F_G}, \boldsymbol{\kappa_G}, \boldsymbol{\rho_G}, \mathcal{T}_F, \mathcal{T}_{GO}, \mathcal{O}$      $\triangleright$ **GRN** predictions, feasibility threshold, grasp obstruction threshold, desired object

1:   $plan \leftarrow []$
2: **if** $F_a(\mathcal{O}) > \mathcal{T}_F$ **then**
3:     $feasibleGrasps = \emptyset$
4:     **for** each $g \in \mathcal{G}$ such that $F_g(\mathcal{O}) > \mathcal{T}_F$ **do**
5:        $feasibleGrasps \leftarrow feasibleGrasps \cup g$
6:     **end for**
7:     $plan.append(Move(\mathcal{O}, feasibleGrasps))$
8:     **return** $plan$
9: **else**
10:     $BlockersPerGraspType \leftarrow [\emptyset \quad \forall g \in \mathcal{G}]$
11:     $freeableGraspTypes \leftarrow \mathcal{G}$
12:     **for** each $g \in \mathcal{G}$ **do**
13:        **if** $\kappa_g(\mathcal{O}) < \mathcal{T}_F$ **then**
14:           $freeableGraspTypes \leftarrow freeableGraspTypes \setminus g$
15:        **else**
16:           **for** $\mathcal{O}' \in \mathcal{N}(\mathcal{O})$ such that $\rho_g(\mathcal{O}, \mathcal{O}') > \mathcal{T}_{GO}$ **do**
17:              **if** $\mathcal{O}'$ is fixed **then**
18:                 $freeableGraspTypes \leftarrow freeableGraspTypes \setminus g$
19:                 **break**
20:              **else**
21:                 $BlockersPerGraspType[g] \leftarrow BlockersPerGraspType[g] \cup \mathcal{O}'$
22:              **end if**
23:           **end for**
24:        **end if**
25:     **end for**
26:     **if** $freeableGraspTypes$ is empty **then**
27:        **return** []
28:     **else**
29:        **for** each $g \in freeableGraspTypes$ **do**
30:           $subplan \leftarrow []$
31:           $graspTypeFreed \leftarrow True$
32:           **for** each $\mathcal{O}' \in BlockersPerGraspType[g]$ **do**
33:              $result \leftarrow GRNPlanner(F_a, \boldsymbol{F_G}, \boldsymbol{\kappa_G}, \boldsymbol{\rho_G}, \mathcal{T}_F, \mathcal{T}_{GO}, \mathcal{O}')$
34:              **if** $result$ is empty **then**
35:                 $graspTypeFreed \leftarrow False$
36:                 **break**
37:              **else**
38:                 $subplan.append(result)$
39:              **end if**
40:           **end for**
41:           **if** $graspTypeFreed$ is True **then**
42:              $plan.append(subplan)$
43:              **return** $plan$
44:           **end if**
45:        **end for**
46:        **if** $graspTypeFreed$ is False **then**
47:           **return** []
48:        **end if**
49:     **end if**
50: **end if**

---

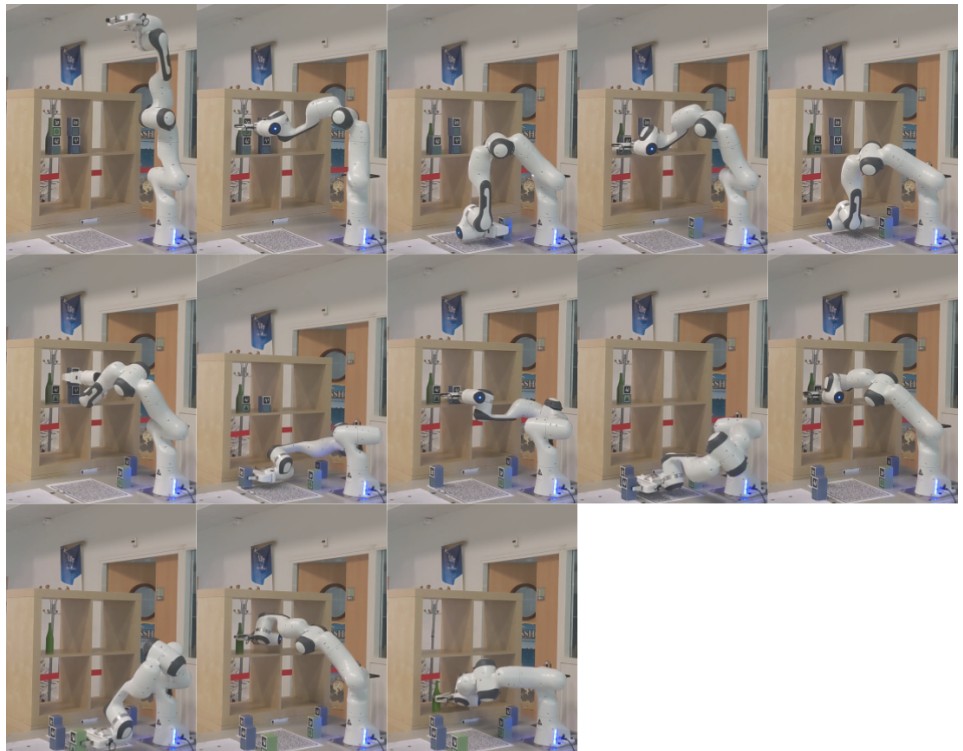

Figure 7: Rollout of the solution found by the **GRN**-based planner on the **Access** problem.

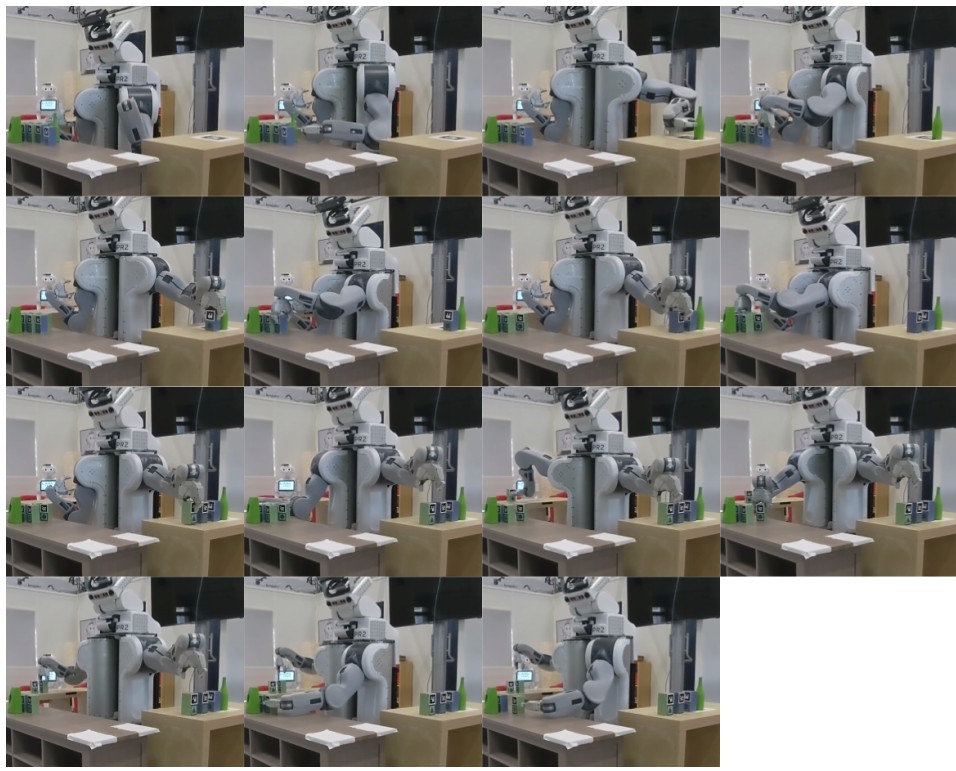

Figure 8: Rollout of the solution found by the **GRN**-based planner on the **Clutter** problem.

## C.2 REAL-WORLD EXPERIMENTS

We test the GRN-based planner in two real-world setups, using a Panda arm and a PR2 robot. We use a perception module based on Overworld (Sarthou, 2023) to identify objects and estimate their poses. The shapes of the object are known and are not subject to detection. The GRN-based planner was able to generate geometrically feasible plans on both the **Access** and **Clutter** problems, which were successfully executed by the robots. Figures 7 and 8 show the rollouts of both executions. In the **Access** problem, the desired task is simply to pick and place the bottle object, the planner has to find the solution plan that needs to move all obstructing objects to get the bottle. In the **Clutter** problem, the goal is to simply pick the object in the middle of the clutter and place it at the center of the table. In the same way, the planner has to find a solution that frees one of the grasp types of the object first. Although these tasks might seem simple, they represent challenging TAMP problem with high combinatorial complexity. Indeed, in both problems, multiple objects not only block access to the goal object, they also block access to each other. The number of geometrically feasible solutions is very limited, requiring the planner to find the specific order in which to move the objects and which grasp to choose at each step. As shown in Figures 7 and 8, the **Access** problem requires 6 Pick-Place actions to be solved, while the **Clutter** problem requires 7 Pick-Place actions to be solved.

## C.3 ADDITIONAL SIMULATION EXPERIMENTS

**Access problem with a large number of objects**. We conduct an additional experiment in simulation to test the scalability of our method to problems requiring long-horizon planning. As in the previous problems, the goal is to move a single object (shown in pink in Figure 9), to which a total of 27 objects block access. This task requires removing at least 18 objects to access the desired one, resulting in a total of 19 necessary Pick-Place actions. Figure 9 shows **GRN**'s predictions on the initial state of this problem. The model accurately predicts that, initially accessing the desired object is infeasible, and that only two blockers can be moved, using a single grasp type each. For simplicity, we show the obtained reasons of infeasibility of the pink object only. However, after the same initial query, our model also predicts the reasons of infeasibility for all the blockers as well. The non-informed TAMP algorithm from Bouhsain et al. (2024) completely fails to solve this problem, even after 2 hours of planning, while the **GRN**-based planner find a feasible solution in under 15s.

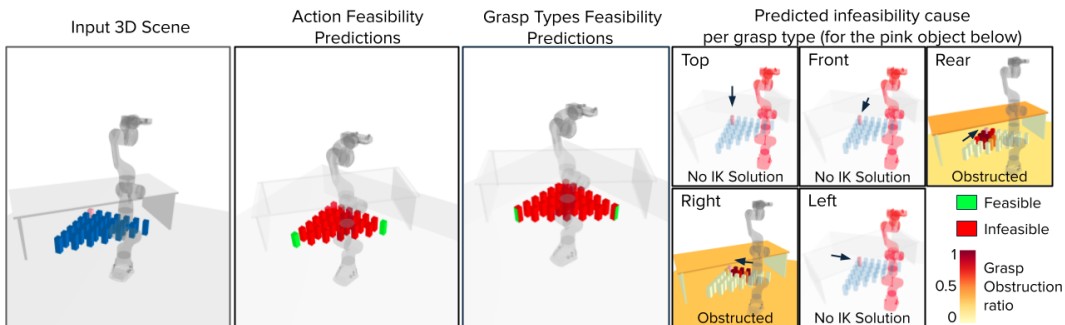

Figure 9: Visualization of **GRN** predictions on the initial state of a larger version of the **Access** problem.

**Inter-robot Handover.** Our proposed approach can also be used to predict the feasibility of handover actions between two robots. Indeed, given the robot-centric nature of our model, we can divide a handover action into two separate Pick-Place actions, and query **GRN** for each robot individually to predict the feasibility of each sub-action. One of the main challenges of handover tasks is to determine where to perform the object exchange. Our model can be used to efficiently find feasible handover positions by evaluating different sampled positions. In order to showcase this ability, we define the **Handover** problem, in which two Panda robots have to perform a handover task for a single object. A wall separates the two robots leaving only 3 "windows" through which the object can be exchanged. We sample a number of potential handover positions that lie in the intersection between the two robots workspaces, and query **GRN** to evaluate their feasibility. As shown in Figure

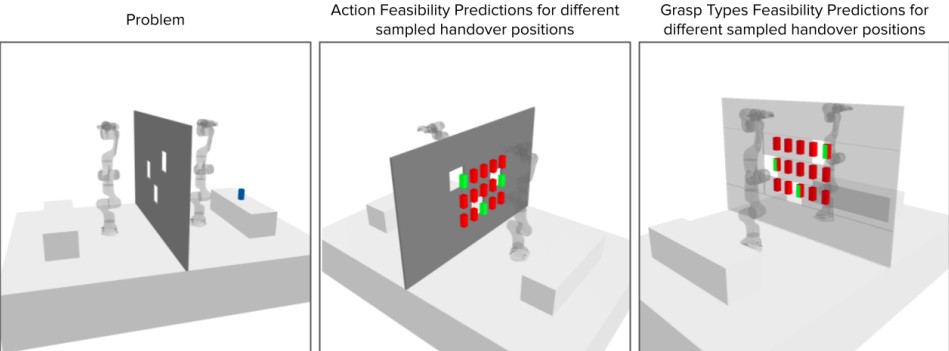

Figure 10: Visualization of **GRN** predictions on different sampled handover positions in the **Handover** problem.

10, our model is able to accurately predict that the wall blocks most of the sampled positions, and that only those in front of the windows are feasible. One key advantage of our method is that all the sampled positions can be evaluated simultaneously by adding a node to the input scene graph for each one, while making sure that proximity edges are not created between these nodes. This enables an efficient search for feasible handover positions, requiring a single query to **GRN**. In contrast, traditional methods need to query the geometric planner for each sampled position independently, resulting in long planning times.

## D INTERPRETABILITY ANALYSIS

To assess the interpretability of our model, we conduct an evaluation using a 3D environment containing two objects of interest for which the action and grasps are initially infeasible, as illustrated in Figure 11. Based on **GRN**'s predicted infeasibility causes, we systematically modify the environment to observe how our model's predictions change in response. This allows us to evaluate the interpretability of our method by showcasing how the predicted reasons of infeasibility explain action and grasp feasibility predictions, and how changing the input scene based on these infeasibility causes affects the outputs of the model.

Figure 11a shows **GRN**'s predictions on the initial 3D environment, and the predicted reasons of infeasibility for the first object of interest. For simplicity, we show infeasibility causes for the *Top* and *Rear* grasp types only. The obtained predictions show that the action involving the object of interest is infeasible, with *Top* grasps being obstructed by the shelf, while *Rear* grasps are obstructed by two movable objects. Figure 11b shows that increasing the height of the shelf removes the obstruction of the *Top* grasp type, with **GRN** predictions indicating that *Top* grasps and the action are now feasible. For the *Rear* grasp type, Figure 11c shows that removing only one of the two obstructing objects does not make the grasp type feasible. Removing the second object as well is necessary to allow access to the object of interest, as shown in Figure 11d.

Regarding the second object of interest, the corresponding reasons of infeasibility are shown in Figure 11e. Due to the object's unreachability by the robot, all grasp types are infeasible due to the absence of IK solutions. Figure 11f shows that decreasing the distance between the object and robot allows reachability, with **GRN** predictions showing that the action as well as the *Top* and *Rear* grasp types become feasible.

This evaluation demonstrates **GRN**'s strong interpretability capabilities. By predicting specific infeasibility causes, namely grasp obstructions and IK infeasibility, our model's feasibility predictions are not only informed, but also interpretable. Moreover, these explanations provide actionable insights into how modifying the environment affects feasibility. Its outputs consistently align with expected changes, sustaining **GRN**'s reliability and coherence. These interpretability features are critical for real-world task and motion planning, where understanding the reasons behind infeasibility is as important as the feasibility predictions themselves.

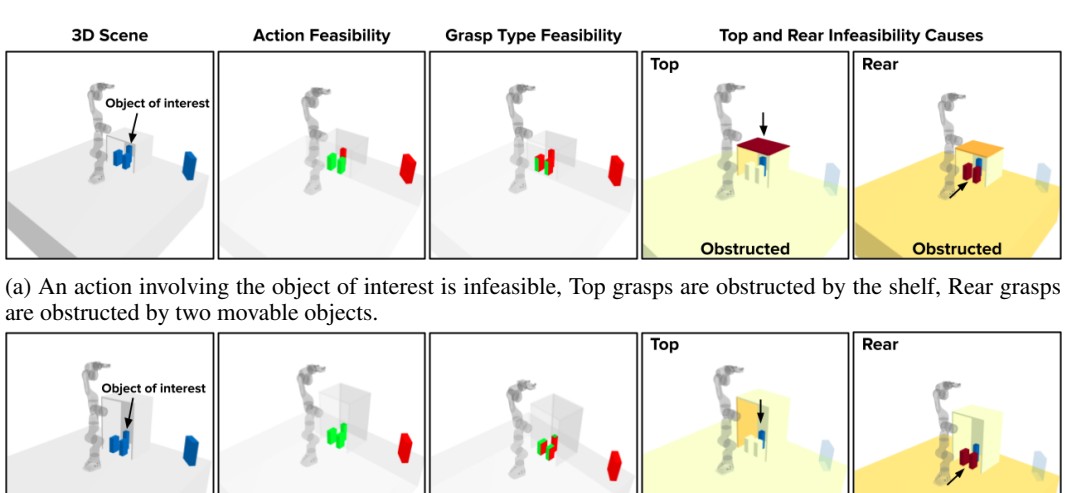

(a) An action involving the object of interest is infeasible, Top grasps are obstructed by the shelf, Rear grasps are obstructed by two movable objects.

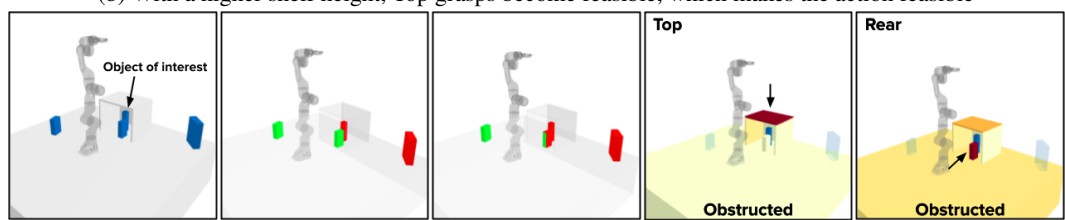

(b) With a higher shelf height, Top grasps become feasible, which makes the action feasible

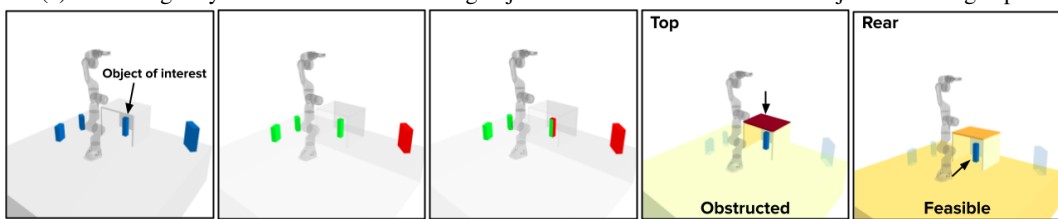

(c) Removing only one of the two obstructing objects does not free access to the object via Rear grasps

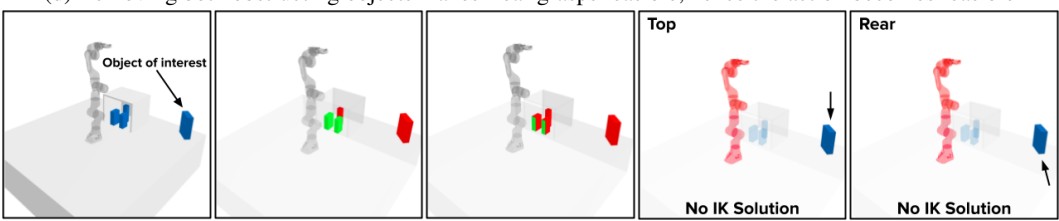

(d) Removing both obstructing objects makes Rear grasps feasible, hence the action becomes feasible

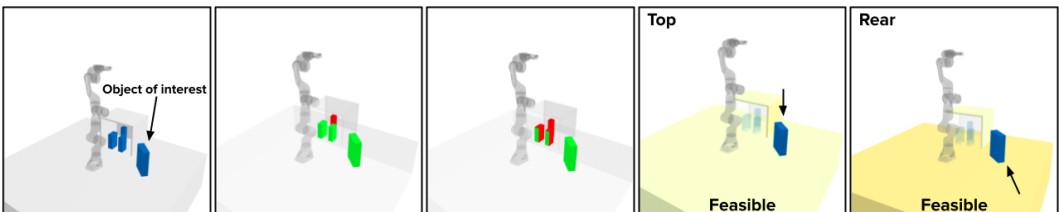

(e) The object of interest is unreachable by the robot, making the action and grasps infeasible due to the absense of IK solutions

(f) Making the object closer to the robot allows reachability, thus the action and grasps become feasible.

Figure 11: Interpretability evaluation of **GRN**'s predictions on an example 3D environments.

# E    ROBUSTNESS TO NOISE

We evaluate the robustness of **GRN** by introducing varying levels of noise to the input data. Specifically, noise is added to the node features, i.e. the dimensions and poses of all objects in the environment. The noise is sampled from a normal distribution with the following parameters: (i) **1cm, 1°** **Noise Level:** Noise with a mean of 0 and a standard deviation of 1 cm for bounding box dimensions and translation coordinates, and 1° for object orientation, (ii) **2cm, 2° Noise Level:** Noise with a mean of 0 and a standard deviation of 2 cm for bounding box dimensions and translation coordinates, and 2° for object orientation. Table 7 presents the performance of GRN on the Panda-3D-4 test set under these noise conditions.

Table 7: Performance of **GRN** on the **Panda-3D-4** test set with different levels of noise.

|            | No Noise         | 1cm, 1°          | 2cm, 2°          |
|------------|------------------|------------------|------------------|
| **Action (F1)** | 0.939       | 0.912            | 0.864            |
| **Grasp (F1)**  | 0.940 ($\pm$ 0.009) | 0.891 ($\pm$ 0.020) | 0.820 ($\pm$ 0.029) |
| **IK (F1)**     | 0.995 ($\pm$ 0.001) | 0.985 ($\pm$ 0.001) | 0.971 ($\pm$ 0.002) |
| **GO (MAE)**    | 0.028 ($\pm$ 0.003) | 0.039 ($\pm$ 0.003) | 0.057 ($\pm$ 0.003) |

While there is a slight decrease in performance as noise levels increase, **GRN** maintains a good accuracy in both action and grasp feasibility predictions. Importantly, even under the highest noise condition (**2 cm, 2°**), our method still achieves a better performance than the one yielded by most baselines on noise-free inputs, shown in Table 1. More precisely, under both noise levels, **GRN** outperforms **MLP**, **DVH** and **F-GCN** on action and grasp feasibility prediction, even when these baselines are given noise-free inputs. Under the **(1cm, 1°)** noise level, our model also provides more accurate predictions than the ones obtained by **AGFPNet** and **F-GAT** under zero noise. Under the highest noise level, **GRN** achieves similar performance to noise-free **F-GAT** and a slightly worse performance than zero-noise **AGFPNet** on action feasibility prediction, but still yields a better performance than both on grasp type feasibility prediction. Also, comparing standard deviations shows that even when noise is applied, **GRN** results in a more consistent prediction quality across grasp types.

Furthermore, we highlight that in addition to a better accuracy under different noise level, our model also provides richer information thanks to our two interpretation modules. Table 7 shows that **GRN** maintains a good performance across noise levels on IK feasibility and Grasp Obstruction (GO) predictions as well. These results emphasize the robustness of GRN to real-world imperfections, while still delivering state-of-the-art performance compared to other methods.

# F    VISUALIZATIONS

In this section, we present a visual comparison between the predictions of our model, GRN, and the ground truth. Figure 12 illustrates GRN's predictions in a test environment from **Panda-3D-4**, while Figure 13 showcases predictions from **Panda-3D-10**. In these figures, action feasibility predictions and ground truth are represented through object colors in the top images: green indicates feasibility for pick or place actions, while red denotes infeasibility. The feasibility of different grasp types is visualized through the coloration of the corresponding object's faces. Finally, the reasons of infeasibility are shown for one of the objects in the environment (in blue), for each grasp type. There are two types of infeasibility causes, IK infeasibility (with the robot depicted in red) and grasp obstructions (where obstructing objects are displayed in full opacity, with their colors reflecting obstruction ratios). These comparisons highlight that GRN accurately predicts action and grasp feasibility for most objects. It also effectively identifies the reasons for infeasibility and estimates grasp obstruction ratios with minimal error. A notable exception occurs in the 10-object environment, where the *Rear* grasp type of one object is incorrectly classified as infeasible. Figure 13 details the predicted infeasibility reasons for this misclassification. A closer examination of the corresponding *Rear* grasp type plot reveals a slight overestimation of the obstruction ratio for a large block, which likely contributes to the error. However, GRN correctly predicts the feasibility of other grasp types for the same object, including the feasible *Top* grasp type, ensuring accurate action feasibility predictions overall.

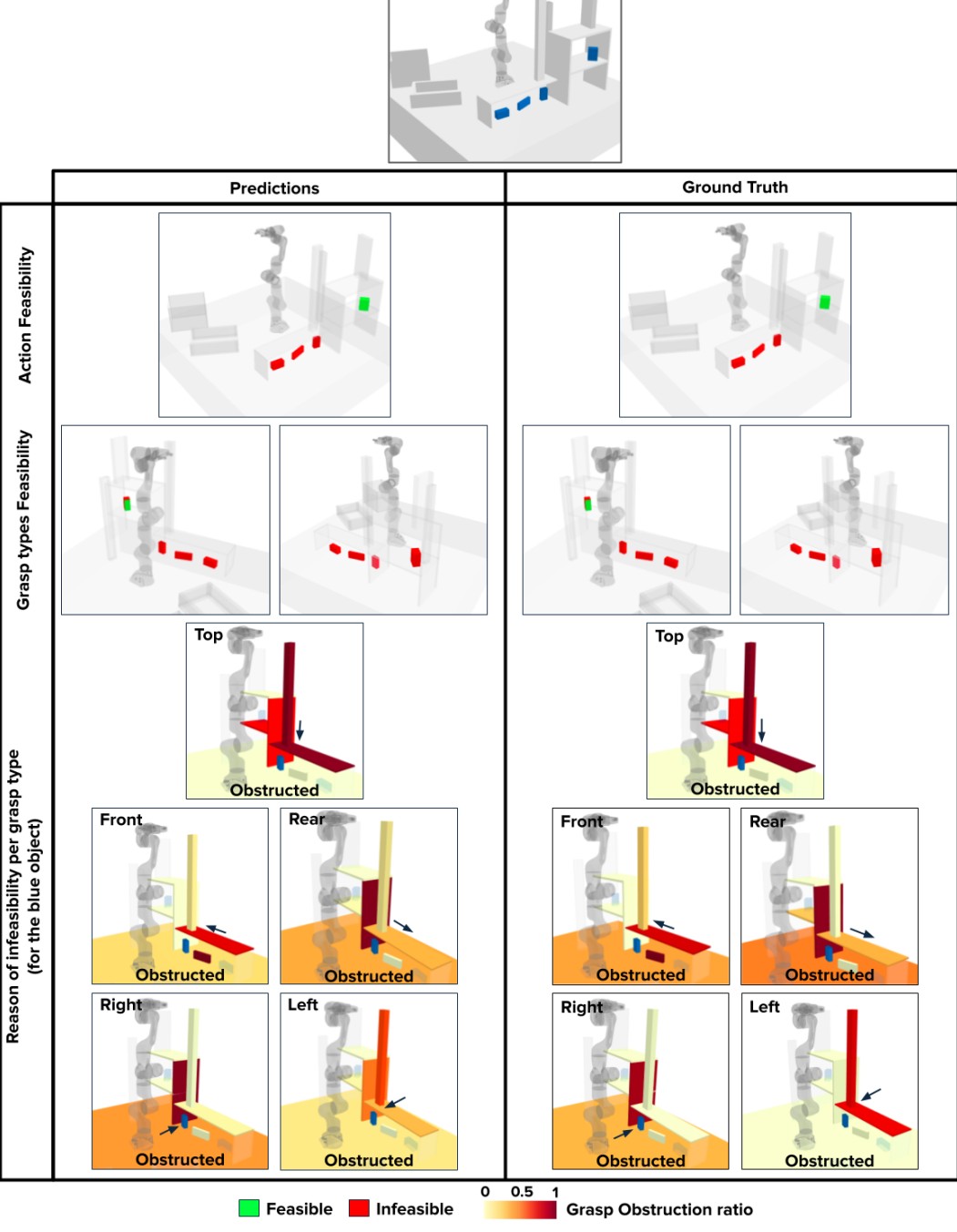

Figure 12: Visualization of **GRN** prediction compared to the ground truth on a test environment from **Panda-3D-4**.

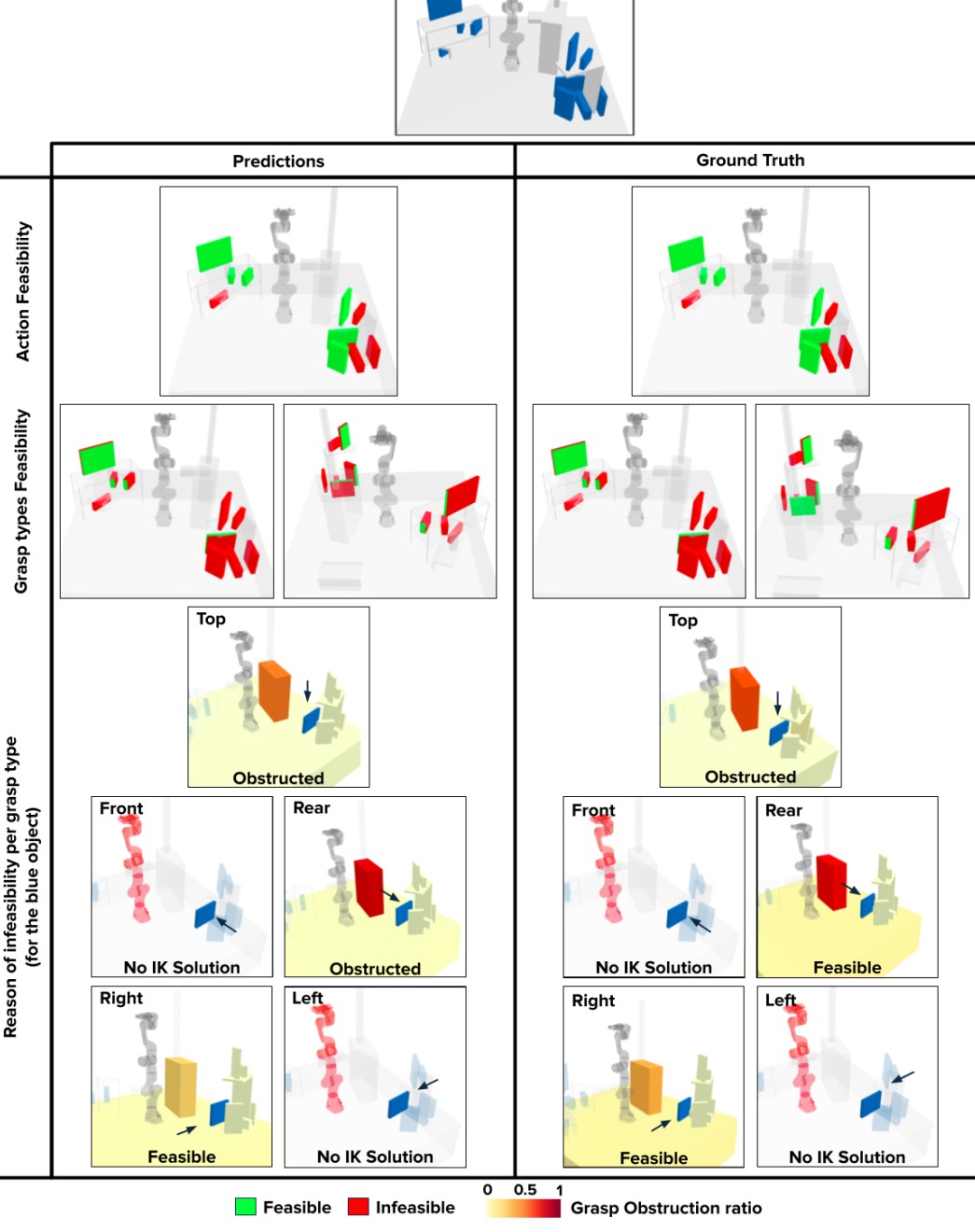

Figure 13: Visualization of **GRN** prediction compared to the ground truth on a test environment from **Panda-3D-10**.

