# OpenReview forum: "Learning Geometric Reasoning Networks For Robot Task And Motion Planning"
_ICLR.cc/2025/Conference — ICLR 2025 Poster_

### Official Review · Reviewer_WQrW · 2024-10-23

**Soundness:** 3
**Presentation:** 2
**Contribution:** 3
**Rating:** 6
**Confidence:** 4

**Summary:**

This paper proposes to use GNN-based methods to predict action and grasp feasibility, which can speed up the geometric planner in task and motion planning. Furthermore, the authors propose to predict the inverse kinematics feasibility and grasp obstruction to improve the interpretability. Through quantitative and qualitative experiments, the authors verify the effectiveness of the proposed approach.

**Strengths:**

1: This paper is well-motivated. The efficiency of task and motion planning is a significant issue and how to improve it is an important research question.

2: Predicting geometric dependencies and feasibility with a graph neural network makes sense to me.

3: This paper contains an extensive experiments section and compares the method to several baselines, which is nice.

4: The paper shows some real-world demos.

**Weaknesses:**

1: My first concern is the limited discussion and comparison to related works in geometric feasibility reasoning [1, 2, 3] and graph neural networks in task and motion planning [4, 5, 6]. In particular, [2] also uses feasibility-related predicates to improve the efficiency of the task planning.

[1]: K. Lin, C. Agia, T. Migimatsu, M. Pavone, and J. Bohg. Text2motion: From natural language instructions to feasible plans. Autonomous Robots, 47(8):1345–1365, 2023.

[2]: Y. Huang, C. Agia, J. Wu, T. Hermans, and J. Bohg. Points2Plans: From Point Clouds to Long-Horizon Plans with Composable Relational Dynamics, ArXiv, 2024.

[3]: C. Agia, T. Migimatsu, J. Wu, and J. Bohg. Stap: Sequencing task-agnostic policies. In 2023 IEEE International Conference on Robotics and Automation (ICRA), pages 7951–7958. IEEE, 2023.

[4]: Planning for Multi-Object Manipulation with Graph Neural Network Relational Classifiers. In IEEE International Conference on Robotics and Automation (ICRA), 2023.

[5]: H. Shi, H. Xu, Z. Huang, Y. Li, and J. Wu. Robocraft: Learning to see, simulate, and shape elastoplastic objects in 3d with graph networks. The International Journal of Robotics Research.

[6]: H. Chen, Y. Niu, K. Hong, S. Liu, Y. Wang, Y. Li, and K. R. Driggs-Campbell, “Predicting object interactions with behavior primitives: An application in stowing tasks,” in 7th Annual Conference on Robot Learning, 2023.

2: The videos and figures for this paper are hard to follow. For example, in Figure 1, I can only understand green represents feasible and red represents feasible. However, there are other colored objects like blue ones. Furthermore, why robots are sometimes grey and sometimes red?

3: This paper makes several strong assumptions including known object shapes and poses, and all objects remain static except the robot grasps them.

**Questions:**

1: how would your proposed approach compare to related works in geometric feasibility reasoning [1, 2, 3] and graph neural networks in task and motion planning [4, 5, 6]?

2: Is there any noise when you estimate the object's shape and pose in the real world? If there is noise, would your proposed approach be robust to the noise?

3: Could you train one more for all robots? Training one model for each robot limits the generalization ability of your proposed system.

4: For the Panda-3D-4 dataset, the MLP achieves a 0.558 F1 score but the Panda-3D-20 achieves a 0.609 F1 score, why does MLP perform better in a more complex dataset? This is weird.

5: In Table 5, why does your proposed approach perform worse than the baseline in the “Access” problem? Why does the baseline always achieve a 100% success rate? I guess your task is too easy.

---

> ### Author Response · Authors · 2024-11-19
> **Response for Reviewer WQrW (Part 1/2)**
>
> We greatly appreciate the reviewer’s comprehensive feedback and address their concerns in detail below.
>
> ---
>
> ### **Comments on raised weaknesses**
>
> **1. Comparison to related works in geometric feasibility reasoning and GNNs in TAMP:**
>
> We thank the reviewer for bringing these works to our attention. We have carefully reviewed the suggested references and included a discussion of these works in the related works section of the revised manuscript.
>
> **2. Figures clarity:**
>
> We appreciate the reviewer’s feedback on the clarity of the figures. In the revised manuscript, we have improved the caption for Figure 1 to make it easier to interpret. Additionally, we have added a detailed explanation and commentary for the figures in Appendix D. To clarify, the visualization figures show the three predictions of GRN: (i) Action feasibility, represented by the object color in the corresponding plots (green for feasible, red for infeasible). (ii) Grasp type feasibility, represented by the coloration of the corresponding object faces. (iii) Reasons of infeasibility: Shown for a single object for clarity., which is depicted in blue (exceptionally in pink in Figure 9). Neighboring objects, shown in full opacity, are color-coded to represent the grasp obstruction ratio. We believe these updates and explanations significantly improve the clarity and comprehensibility of the figures and welcome any further suggestions.
>
> **3. Scene knowledge and static environments assumptions:**
>
> We acknowledge the reviewer’s concern regarding the assumptions of known object shapes, poses, and static scenes. However, such assumptions are standard in offline TAMP research, as demonstrated in both traditional and learning-based works [7, 8, 9, 10, 11]. These works typically decouple the perception and planning problems, focusing on solving the manipulation planning task with fully defined scenes. This decoupling is a common practice in offline manipulation planning to allow methods to focus on geometric reasoning. Our approach aligns with these standards, making the assumptions consistent with the existing literature.
>
> ---
>
> ### **Answers to raised questions**
>
> **1. Comparison to geometric feasibility reasoning and GNNs in TAMP:**
>
> [1] and [2] propose learning-based methods for TAMP using large language models. Unlike [1], which relies solely on the probability of feasibility of learned skill primitives, our method aligns more closely with [2] by explaining the reasons behind infeasibility. This interpretability allows downstream planners to bypass infeasible actions more effectively. Points2Plans ([2]) trains relational dynamics models for single-step transitions, whereas GRN simplifies feasibility checks into interpretable modules, making it straightforward to train and integrate with traditional TAMP planners. GRN could also serve as an independent feasibility checker for [1] and [2].
>
> [3] uses RL to learn task-agnostic robot skills and verify their feasibility for manipulation tasks. While this method requires a separate model per skill and a single query per action, GRN learns a unified model for both pick and place actions, predicting feasibility for all objects simultaneously. Additionally, GRN’s interpretability mechanisms provide insights into infeasibility, enhancing decision-making.
>
> [4] and [6] utilize GNNs for different aspects of TAMP. [4] learns inter-robot relations to verify subgoal satisfaction with a scene graph representation similar to ours. [6] leverages GNNs for dynamic object interactions in stowing tasks. However, neither focuses specifically on action and grasp feasibility prediction. GRN could be integrated into these methods to predict feasibility-related inter-robot relations or evaluate the feasibility of stowing action sequences.
>
> [5] addresses shaping deformable objects, a task unrelated to action and grasp feasibility prediction. As such, it does not align closely with our focus.
>
> In summary, GRN distinguishes itself through its focus on feasibility prediction, interpretability, and modularity, making it a complementary tool for integration into various TAMP frameworks. The mentioned works, however, tackle non-prehensile actions as well, which is a limitation of our method we aim to address in future work.
>
> **2. Robustness to noise in real-world experiments:**
>
> In our real-world experiments, the estimated object poses are indeed noisy, while the object shapes are exact as they are obtained through object recognition, primarily using tags. Despite this, the observed success of the GRN-based planner on real-world tasks suggests that our approach is robust to relatively small noise in pose estimation.
>
> It is important to note that GRN predicts the success of a sampling-based geometric planner (defined as feasibility). While larger noise levels would reduce GRN’s performance, they would similarly undermine the success of the geometric planner itself, as both rely on accurate environment representations.

---

> > ### Author Response · Authors · 2024-11-19
> > **Response for Reviewer WQrW (Part 2/2)**
> >
> > **3. Training one model for all robots:**
> >
> > Our current implementation does not support training a single model for all robots. This limitation is common among existing methods for feasibility prediction, as kinematics and collision models can vary significantly across different robots. Training a reliable unified model would require incorporating a representation of the kinematics and collision model for each robot, which can be complex to encode effectively. That said, this is an interesting direction for future work, and we aim to explore ways to address this challenge in subsequent research.
> >
> > **4. MLP generalizability to a larger number of objects:**
> >
> > We appreciate the reviewer’s observation. In our problem, IK feasibility prediction is the simplest task (cf. Table 3), with a simple MLP (IK module) achieving an F1 score exceeding 99%. Environments with more objects naturally lead to an increase in infeasibility cases due to unreachability, which depends solely on the object’s position and not its neighborhood. This explains the seemingly good generalization of the MLP baseline in larger environments, as it performs well at predicting unreachability.
> >
> > However, the MLP baseline does not account for neighboring obstacles and performs poorly in grasp obstruction cases, where its predictions are effectively random. Comparing confusion matrices for Panda-3D-20 below, the MLP baseline yields a high number of false positives. In contrast, GRN incorporates both IK and GO considerations, leading to more balanced and accurate predictions, with a notable improvement in F1 scores 0.2 higher for action feasibility and 0.3 higher for grasp feasibility on Panda-3D-20 w.r.t MLP.
> >
> > ---
> >
> > **MLP Confusion Matrix on Panda-3D-20**
> >
> > | Actual / Predicted | Infeasible | Feasible |
> > |---------------------|------------|----------|
> > | **Infeasible**      | 65129      | 23065    |
> > | **Feasible**        | 4679       | 27127    |
> >
> > ---
> >
> > **GRN Confusion Matrix on Panda-3D-20**
> >
> > | Actual / Predicted | Infeasible | Feasible |
> > |---------------------|------------|----------|
> > | **Infeasible**      | 85956      | 2288     |
> > | **Feasible**        | 3856       | 27910    |
> >
> > ---
> >
> > **5. Performance on the Access problem:**
> >
> > The baseline TAMP planner is a complete, closed-loop algorithm that iteratively evaluates multiple task plans until a feasible one is found. In contrast, our GRN-based planner is a one-shot open-loop algorithm: if the initial plan generated using GRN’s predictions is infeasible, the planning process terminates, and no alternative plans are considered. In one instance of the Access problem, GRN produces an incorrect prediction, resulting in a geometrically infeasible plan, and thus a planning failure.
> >
> > The considered problems are not particularly easy (see Appendix C.2). However, we deliberately limit the number of objects to ensure that the baseline planner can solve the problems in a reasonable amount of time, allowing for a fair comparison of planning time. To further demonstrate the capabilities of GRN, we include an additional experiment in Appendix C.3, featuring a more challenging 28-object Access problem. The baseline planner fails completely on this task, while the GRN-based planner solves it in under 15 seconds.
> >
> > ---
> > **References**
> > [1]: K. Lin, C. Agia, T. Migimatsu, M. Pavone, and J. Bohg. Text2motion: From natural language instructions to feasible plans. Autonomous Robots, 2023.
> >
> > [2]: Y. Huang, C. Agia, J. Wu, T. Hermans, and J. Bohg. Points2Plans: From Point Clouds to Long-Horizon Plans with Composable Relational Dynamics, ArXiv, 2024.
> >
> > [3]: C. Agia, T. Migimatsu, J. Wu, and J. Bohg. Stap: Sequencing task-agnostic policies. In 2023 IEEE International Conference on Robotics and Automation, ICRA 2023.
> >
> > [4]: Planning for Multi-Object Manipulation with Graph Neural Network Relational Classifiers. In IEEE International Conference on Robotics and Automation, ICRA 2023.
> >
> > [5]: H. Shi, H. Xu, Z. Huang, Y. Li, and J. Wu. Robocraft: Learning to see, simulate, and shape elastoplastic objects in 3d with graph networks. IJRR 2024.
> >
> > [6]: H. Chen, Y. Niu, K. Hong, S. Liu, Y. Wang, Y. Li, and K. R. Driggs-Campbell, “Predicting object interactions with behavior primitives: An application in stowing tasks,” CoRL 2023.
> >
> > [7] Garrett, Caelan Reed, et al. "Integrated task and motion planning." Annual review of control, robotics, and autonomous systems 4.1 (2021).
> >
> > [8] Garrett, Caelan Reed, Tomas Lozano-Perez, and Leslie Pack Kaelbling. "Ffrob: Leveraging symbolic planning for efficient task and motion planning." IJRR 2018.
> >
> > [9] Wells, Andrew M., et al. "Learning feasibility for task and motion planning in tabletop environments." RAL 2019.
> >
> > [10] Driess, Danny, et al. "Deep visual heuristics: Learning feasibility of mixed-integer programs for manipulation planning." ICRA 2020.
> >
> > [11] Bouhsain, Smail Ait, et al . "Simultaneous Action and Grasp Feasibility Prediction for Task and Motion Planning through Multi-Task Learning." IROS 2023.
> >
> > ---

---

> > > ### Comment · Reviewer_WQrW · 2024-11-26
> > >
> > > Thank you for the detailed responses. Most of my concerns have been addressed, and I have updated my rating accordingly.
> > >
> > > However, I still have concerns about the generalization of this work to unstructured, uncertain, real-world environments due to the strong assumptions about known object shape and pose. While the current real-world experiments appear to perform well, they do so only under limited noise and in a simplified experimental setup. Therefore, I cannot further improve my rating.

---

> > > > ### Author Response · Authors · 2024-11-29
> > > >
> > > > We sincerely thank you for your valuable feedback and your reassessment of our work.
> > > >
> > > > To address your remaining concerns regarding generalization to unstructured, noisy environments, we conducted an additional experiment to evaluate GRN's robustness to input noise. This experiment, detailed in Appendix E of the newly revised manuscript, introduces varying levels of Gaussian noise to objects' poses and bounding box dimensions during inference. Results on the Panda-3D-4 test set demonstrate that GRN retains a good performance under different noise levels. Also, even under the highest noise level, our model still outperforms baselines evaluated on noise-free data, especially on grasp feasibility prediction.
> > > >
> > > > We highlight that in addition to state-of-art performance on action and grasp feasibility prediction, GRN provides richer information thanks to its two interpretation mechanisms. We also provide an analysis of the impact of noise on IK feasibility and grasp obstruction predictions, which remain reliable for guiding downstream planning tasks.
> > > >
> > > > **Performance of GRN on the Panda-3D-4 test set under different levels of noise**
> > > > | Noise Level   	| Action (F1) | Grasp (F1)       	| IK (F1)       	| GO (MAE)       	|
> > > > |--------------------|-------------|-----------------------|--------------------|---------------------|
> > > > | No Noise      	| 0.939   	| 0.940 (± 0.009)  	| 0.995 (± 0.001)   | 0.028 (± 0.003)	|
> > > > | 1 cm, 1°      	| 0.912   	| 0.891 (± 0.020)  	| 0.985 (± 0.001)   | 0.039 (± 0.003)	|
> > > > | 2 cm, 2°      	| 0.864   	| 0.820 (± 0.029)  	| 0.971 (± 0.002)   | 0.057 (± 0.003)	|
> > > >
> > > > We hope these additional results and analysis further address your concerns about GRN's robustness to noise. We appreciate your constructive comments and remain open to any further discussion.

---

> > > > > ### Author Response · Authors · 2024-12-02
> > > > >
> > > > > Dear Reviewer,
> > > > >
> > > > > As the discussion phase is nearing its conclusion, we kindly request your feedback and reassessment based on the revisions and additional experiments we have provided, particularly the robustness analysis under noisy inputs in Appendix E. We hope these additions address your remaining concerns and demonstrate the broader applicability and generalization of our approach.
> > > > >
> > > > > If you have any further questions, we would be more than happy to address them. Thank you once again for your time and valuable contributions to this review process.

---

> ### Comment · Area_Chair_Ecfi · 2024-11-25
>
> Dear Reviewer,
>
> Please provide feedback to the authors before the end of the discussion period, and in case of additional concerns, give them a chance to respond.
>
> Timeline: As a reminder, the review timeline is as follows:
>
> November 26: Last day for reviewers to ask questions to authors.
>
> November 27: Last day for authors to respond to reviewers.

---

### Official Review · Reviewer_FvER · 2024-10-31

**Soundness:** 3
**Presentation:** 3
**Contribution:** 2
**Rating:** 8
**Confidence:** 3

**Summary:**

The paper introduces Geometric Reasoning Networks (GRN), a GNN model for predicting action and grasp feasibility. Technical Innovations of the paper includes:
- Novel scene representation using directed graphs
- Edge-Enhanced Graph Attention Network (EGAT) that leverages both node and edge features
- Ability to explain why actions are infeasible, enabling more efficient planning

**Strengths:**

1. Outperforms state-of-the-art methods in both action and grasp feasibility prediction
2. Lower inference time compared to traditional geometric planning
3. Generalization to more complex environments and different robot types
4. Writing is very clear and detailed

**Weaknesses:**

1. Some details for the experiment setting needed to be discussed further to correctly evaluate the approach. See the questions.

**Questions:**

1a. How imbalanced is the dataset? Is it nearly composed of 50% feasible cases and 50% infeasible cases? It would be appreciated if some ratio of the dataset could be shared as a table.

1b. Since RRT is a random algorithm, I think it doesn't necessarily mean one task is truly infeasible, if RRT couldn't generate a plan. What is the definition of the feasibility in this paper's scope?

2. Regarding training, why is pre-trained needed before jointly fine-tuning? Was there any training instability if we directly train all the modules?

3. The generalization experiment is interesting. What I'm expecting is that the graph representation should be powerful enough to achieve a significantly better generalization. However, it seems that the MLP method is also good enough to generalize to 20 obstacles. Is there any specific reason why this is the case? Or are we also facing the imbalanced dataset issue here? Sampling a feasible case for 20 obstacles sounds challenging, right?

4. I am not very familiar with TAMP problem. Just as a discussion, in general, when would a learning-based feasibility checker be preferred compared to a learning-based end-to-end planner, that directly generates a plan? And when vice versa?

---

> ### Author Response · Authors · 2024-11-19
> **Response for Reviewer FvER**
>
> We sincerely value the reviewer’s input and have addressed their questions in the sections below:
>
> **1. Dataset Imbalance:**
>
> We have included a detailed description of the dataset distribution in Appendix B for the Panda-3D-4 and PR2-3D-4 datasets. Action feasibility labels are well balanced, with roughly equal numbers of feasible and infeasible cases. Grasp feasibility labels, however, are imbalanced, with a higher proportion of infeasible cases. Importantly, the reasons for infeasibility are balanced between IK infeasibility and grasp obstructions in the Panda-3D-4 dataset, while the PR2-3D-4 dataset contains more IK infeasibility cases. Our proposed data augmentation methods help mitigate these imbalances. Motion planning infeasibility cases are rare across both datasets.
>
> **2. RRT and definition of feasibility:**
>
> We agree with the reviewer’s observation regarding RRT. However, since RRT is probabilistically complete, running RRT for an infinite amount of time would be needed to guarantee that a plan will be found if it exists. In practice, we set a timeout during data annotation, and consider an action as infeasible if the geometric planner fails to find a solution within this timeout. Thus, feasibility is defined as the success of the chosen geometric planner in finding a solution within a user-defined timeout.
>
> It is worth noting that in our datasets, as shown in our updated Appendix B, most infeasibility cases arise from inverse kinematics (IK) constraints or grasp obstructions (GO), rather than from motion planning (MP) failures.
>
> **3. Pros of pre-training modules independently:**
>
> We observed no instability when training all modules jointly. However, pre-training ensures that each module effectively learns its specific task given its designated inputs. Without pre-training, jointly training all modules leads to convergence to spurious local optima due to the greedy nature of weight optimization (cf. ablation study). Pre-training helps mitigate this issue by providing a strong initialization, which ensures more effective fine-tuning during joint training.
>
> **4. MLP’s generalizability to larger environments:**
>
> We appreciate the reviewer’s observation. The simplest task in our problem is IK feasibility prediction, which even a simple MLP achieves with an F1 score exceeding 99% (cf. ablation study). Generating environments with more objects increases cases where infeasibility is due to unreachability, which depends solely on the object’s position rather than its neighborhood. This explains the seemingly good generalization of the MLP baseline in larger environments, as it excels at predicting unreachability.
>
> However, the MLP baseline does not account for neighboring obstacles and struggles in cases of grasp obstruction, where its predictions are no better than random. Comparing confusion matrices obtained on Panda-3D-20 shown below, the MLP baseline frequently predicts actions and grasps as feasible, resulting in a high number of false positives. GRN, by contrast, incorporates both IK and GO considerations, producing more balanced and informed predictions, which explains the significant difference in F1 score of 0.2 (resp. 0.3), for action (resp. grasp) feasibility prediction.
>
> ---
>
> **MLP Confusion Matrix on Panda-3D-20**
>
> | Actual / Predicted | Infeasible | Feasible |
> |---------------------|------------|----------|
> | **Infeasible**      | 65129      | 23065    |
> | **Feasible**        | 4679       | 27127    |
>
> ---
>
> **GRN Confusion Matrix on Panda-3D-20**
>
> | Actual / Predicted | Infeasible | Feasible |
> |---------------------|------------|----------|
> | **Infeasible**      | 85956      | 2288     |
> | **Feasible**        | 3856       | 27910    |
>
> ---
>
> **5. Learned feasibility checker Vs. Learned end-to-end planner:**
>
> Our solution based on a learned feasibility checker aims to limit the calls to a costly geometric planner. It ensures completeness, generalizability to unseen environments and diverse robots, interpretability for geometric reasoning, and safety through traditional collision-free motion planning, all with lower data generation costs as only binary feasibility labels (and GO ratios) are needed. But it still requires a geometric planner to plan motions for actions predicted as feasible, increasing planning time. In contrast, a learned end-to-end planner could be faster, as it completely avoids geometric queries. However, it sacrifices key requirements such as safety since collision-free motions are not guaranteed, it struggles with generalization to new environments and high-DOF robots, and demands higher data generation costs, requiring full motion plans for training.
> In offline manipulation planning where completeness, safety and generalizability are desirable, learned feasibility checkers are therefore a preferable option.
>
> ---

---

> ### Comment · Area_Chair_Ecfi · 2024-11-25
>
> Dear Reviewer,
>
> Please provide feedback to the authors before the end of the discussion period, and in case of additional concerns, give them a chance to respond.
>
> Timeline: As a reminder, the review timeline is as follows:
>
> November 26: Last day for reviewers to ask questions to authors.
>
> November 27: Last day for authors to respond to reviewers.

---

> > ### Comment · Reviewer_FvER · 2024-11-25
> > **Score Raised to 8**
> >
> > Thank you. I think all my concerns have been addressed, and the additional table helps explain when MLP could fail and GNN does well. While the novelty might be incremental, I don't see any reason to reject it.

---

> > > ### Author Response · Authors · 2024-11-26
> > >
> > > We sincerely thank you for your updated assessment. We are glad that our revisions and additional clarifications addressed your concerns effectively. We greatly appreciate your constructive feedback, which has been invaluable in improving the clarity and depth of our work.

---

### Official Review · Reviewer_edpJ · 2024-11-04

**Soundness:** 3
**Presentation:** 3
**Contribution:** 3
**Rating:** 6
**Confidence:** 3

**Summary:**

The paper presents a graph neural network model for efficient Task and Motion Planning (TAMP). It uses a scene graph to represent 3D environments, incorporating Inverse Kinematics (IK) and Grasp Obstruction (GO) modules. Experiments show its superiority over state-of-the-art methods in accuracy, generalization, and speed. It reduces the reliance on computationally expensive geometric planners while remaining robust across diverse robots and complex scenes.

**Strengths:**

1. It incorporates explainable modules within the GNN, such as inverse kinematics and grasp obstruction estimations, to predict feasibility. This interpretability enhances decision-making by providing insights into why certain actions are infeasible.

2. Experiments demonstrate that GRN significantly lowers computational costs compared to traditional geometric planners, making it a more efficient solution for real-time TAMP applications.

**Weaknesses:**

1. The method's reliance on simplified bounding box representations and discrete grasp feasibility may limit its effectiveness in real-world TAMP problems, especially in dynamic, cluttered environments with irregular objects or sensor noise.

2. The method only assesses discrete feasibility without motion planning, restricting its ability to handle complex cases where motion feasibility [1] is crucial.

[1] Scaling Multi-Modal Planning: Using Experience and Informing Discrete Search

**Questions:**

1. How would the performance of GRN be affected if the Inverse Kinematics (IK) and Grasp Obstruction (GO) modules were replaced with traditional methods, such as standard inverse kinematics solvers or simpler distance-based obstruction checks? Specifically, how would this change impact prediction accuracy, interpretability, and computational efficiency in various environments?

2. What is the maximum level of environmental complexity that GRN can handle effectively? For example, how many fixed nodes or objects can it support while maintaining accuracy and efficiency? Could GRN scale to environments with thousands of nodes, akin to point cloud representations, and still perform well in highly cluttered scenes?

---

> ### Author Response · Authors · 2024-11-19
> **Response for Reviewer edpJ**
>
> We sincerely thank the reviewer for their valuable feedback and address their concerns below:
>
> ---
>
> ### **Comments on raised weaknesses**
>
> **1. Reliance on simplified bounding box representations and discrete grasp feasibility:**
>
> We acknowledge the reviewer’s concern and emphasize that our use of bounding box representations and discrete grasp feasibility strikes a balance between computational efficiency and representational capability. These simplifications have proven effective in predicting action and grasp feasibility for everyday objects [1][2][3].
>
> For more complex objects, finer representations—such as multiple bounding boxes, alternative primitives (cylinders, spheres), or advanced object shape encoders—could be explored. Similarly, continuous grasp feasibility predictions could be achieved with additional data annotation and training efforts, allowing for greater adaptability. We aim to investigate these directions in future work to enhance GRN’s applicability to more challenging scenarios.
>
> **2. Assessing motion planning feasibility:**
>
> We clarify that motion planning feasibility is considered during dataset annotation and training. However, in the generated datasets, such cases are relatively rare (cf. Appendix B in the updated manuscript) and are therefore under-represented in our experiments.
>
> From our experience in TAMP and observations from the generated datasets, the vast majority of infeasibility cases arise from inverse kinematics or grasp obstructions. These dominate failure modes in practical scenarios, making our focus on these aspects both relevant and impactful.
>
> That said, we recognize the importance of addressing complex cases where motion feasibility is a limiting factor. As mentioned in the future work section, we aim to develop a new interpretation module for motion feasibility detection and explore methods to estimate the configuration space’s connectivity using our graph representation, further enhancing GRN’s capabilities in these scenarios.
>
> ---
>
> ### **Answers to raised questions**
>
> **1. Using traditional methods as IK and GO modules:**
>
> Replacing the IK and GO modules with traditional methods, such as standard IK solvers or distance-based obstruction checks, would significantly impact computational efficiency. On average, for a pick action, IK computation and collision checking take approximately 250 ms (includes computations for all considered grasps), making these methods substantially slower than our learned modules which take 0.5ms each (cf. Appendix A in the updated manuscript).
>
> In terms of accuracy and interpretability, we observed that the pretrained GNN (before fine-tuning) using ground truth IK feasibility and grasp obstruction values achieves F1 scores of 0.95 and 0.96 for action and grasp feasibility predictions, respectively. However, we believe that the slight improvement in accuracy does not justify the significant computational overhead introduced by traditional methods, especially in time-sensitive TAMP applications.
>
> **2. Maximum environmental complexity:**
>
> While it is difficult to quantify an exact maximum level of environmental complexity that GRN can handle, we believe it could scale effectively to environments with a high number of objects. This scalability would stem from the localized nature of feasibility predictions: only the distance-based neighborhood of a specific object is considered for each prediction. Even in environments containing thousands of objects, the number of objects in any single neighborhood is inherently limited, ensuring manageable computational requirements and maintaining high prediction quality.
>
> In scenarios involving point clouds, the environment can be decomposed into separate objects, support surfaces, and obstacles. This decomposition allows GRN to operate on a structured representation rather than directly on the raw point cloud. By leveraging this structured approach, GRN could preserve prediction accuracy and efficiency, even in highly cluttered and complex scenes.
>
> ---
>
> **References**
> [1] Wells, Andrew M., et al. "Learning feasibility for task and motion planning in tabletop environments." IEEE robotics and automation letters 4.2 (2019): 1255-1262.
>
> [2] Bouhsain, Smail Ait,et al . "Extending Task and Motion Planning with Feasibility Prediction: Towards Multi-Robot Manipulation Planning of Realistic Objects." 2024 IEEE/RSJ International Conference on Intelligent Robots and Systems (IROS). IEEE, 2024.
>
> [3] Khodeir, Mohamed, et al. "Policy-guided lazy search with feedback for task and motion planning." 2023 IEEE International Conference on Robotics and Automation (ICRA). IEEE, 2023.

---

> > ### Comment · Reviewer_edpJ · 2024-11-23
> > **Response to authors**
> >
> > Thanks for your response, and I appreciate the explanation regarding IK and collision-checking times. However, I find the reported 250ms for traditional methods somewhat high compared to benchmarks like cuRobo [1], which achieves collision-free computation in about 5ms for batch size 10 with GPU acceleration. Could you clarify the following points to provide more context?
> >
> > 1. Batch Size: What batch size was used for both MLP-based IK and traditional IK computations?
> >
> > 2. Parallelization: Were traditional IK computations or collision checking parallelized or GPU-accelerated?
> >
> > 3. Solver Details: What IK solver and collision-checking methods were used?
> >
> > Addressing these points would help align the benchmarks and clarify the comparison. Thank you.
> >
> > [1] cuRobo: Parallelized Collision-Free Minimum-Jerk Robot Motion Generation

---

> > > ### Author Response · Authors · 2024-11-25
> > > **Reponse to follow-up comment from Reviewer edpJ**
> > >
> > > Thank you very much for your response and the opportunity to clarify your remaining concerns. Below, we address the raised points:
> > >
> > > **Traditional solver details:**
> > >
> > > We use MoveIt Task Constructor (MTC) [1] as our geometric planner, which leverages the widely used KDL plugin [2] for inverse kinematics and FCL library [3] for collision checking. These methods are neither parallelized nor GPU-accelerated in our implementation. For each object, we first uniformly sample a set of end-effector grasps based on object size. For each sampled grasp, the KDL solver computes up to 8 inverse kinematics (IK) solutions. Each IK solution is then checked for collisions using the FCL collision checker. We revised Appendix B to make these details clearer.
> > >
> > > **Batch size and computation time:**
> > >
> > > In the Panda-3D-4 dataset, an average of 150 IK solutions is computed and tested for collisions per movable object. This can be considered the batch size for our traditional methods. The reported 250 ms includes all computations for this batch size.
> > >
> > > We agree that recent GPU-accelerated methods like cuRobo [4] can significantly reduce computation times. For example, cuRobo reports 14.7 ms for a batch size of 100, implying it could handle our average batch size of 150 in approximately 22 ms, which is 11 times faster than our chosen traditional methods.
> > >
> > > **Comparison to our IK feasibility and GO modules:**
> > >
> > > cuRobo and other traditional methods verify IK feasibility by actually computing IK solutions and checking for collisions. Many of these solutions may be infeasible, resulting in computation time unnecessarily wasted on infeasible solutions. Our method avoids this inefficiency by predicting which grasps (end-effector poses) are promising, allowing traditional geometric planners to focus only on potentially feasible solutions.
> > >
> > > Furthermore, our IK feasibility and GO modules, in contrast to traditional methods, take as input object features (e.g., dimensions, poses) rather than individual sampled grasps (i.e end-effector poses), and simultaneously predict IK feasibility and grasp obstruction ratios for 5 grasp types (i.e sets of grasps associated with each face of the object's bounding box). This design enables our modules to reason over batches of objects instead of batches of grasps.
> > >
> > > While cuRobo significantly reduces computation time thanks to GPU-acceleration, our IK feasibility and GO modules compute predictions for 4 movable objects in under 1 ms (cf. Appendix A). Based on the previous observations, running cuRobo on the same environment with 150 IK computations per object, would require 88 ms (22 ms × 4). Thus, our IK feasibility and GO modules are **88 times faster**.
> > >
> > > In summary, while GPU-accelerated methods like cuRobo offer significant improvements over other traditional solvers, GRN operates at a fraction of their computational cost while predicting feasibility directly from object features without sampling individual grasps. Our method offers a pre-selection step for deciding which grasps traditional methods should focus on.
> > >
> > > ---
> > >
> > > [1] Görner, Michael, et al. "Moveit! task constructor for task-level motion planning." 2019 International Conference on Robotics and Automation (ICRA). IEEE, 2019.
> > >
> > > [2] https://www.orocos.org/kdl.html
> > >
> > > [3] Pan, Jia, Sachin Chitta, and Dinesh Manocha. "FCL: A general purpose library for collision and proximity queries." 2012 IEEE International Conference on Robotics and Automation. IEEE, 2012.
> > >
> > > [4] Sundaralingam, Balakumar, et al. "CuRobo: Parallelized collision-free minimum-jerk robot motion generation." arXiv preprint arXiv:2310.17274 (2023).

---

> > > > ### Comment · Reviewer_edpJ · 2024-11-25
> > > > **Thank you for the response**
> > > >
> > > > Thank you for providing details about the experimental setup. While I appreciate the work's effort to replace the graph search module in task planning with GNN and to accelerate feasibility checks using neural IK and collision modules (correct me if my summary is not correct), this approach does not sufficiently address my concerns about simplified setup to give a higher score. However, I hope other reviewers could actively engage in this discussion to help clarify the issue further.
> > > >
> > > > On a broader note, the lack of reviewer engagement has been highlighted as a concern in this year’s ICLR. I hope more discussion to ensure a constructive review process.

---

> > > > > ### Author Response · Authors · 2024-11-25
> > > > >
> > > > > We sincerely thank you for your prompt response and engagement. We greatly value your thoughtful feedback and understand your remaining reservations.
> > > > >
> > > > > To clarify, while this paper aims to showcase the interpretability of GRN and the possibilities it offers by proposing a GRN-based planner that circumvents the need for a task planner, the ultimate goal of our approach is not to replace the graph search module in task planning. Instead, our method is designed to provide search heuristics to this module that accelerate the planning process while maintaining completeness.
> > > > >
> > > > > In this work, we focus on demonstrating the efficiency, generalizability, and interpretability of GRN as an independent neural feasibility checker that can be seamlessly integrated into traditional task and motion planning algorithms. Furthermore, we are actively working on a full integration of GRN into a TAMP algorithm that we plan to submit as a future contribution to a more robotics-oriented conference.
> > > > >
> > > > > We hope this response, along with constructive discussions with other reviewers, will help address your remaining concerns. We deeply appreciate your continued engagement and contribution to the review process.

---

> > > > > > ### Author Response · Authors · 2024-11-29
> > > > > >
> > > > > > Dear reviewer edpJ,
> > > > > >
> > > > > > In order to further address your concerns regarding the effectiveness of our method on noisy input data, we conducted an additional experiment evaluating GRN’s robustness to sensor noise, as detailed in Appendix E of the latest manuscript. This experiment introduces varying levels of Gaussian noise to objects' poses and bounding box dimensions during inference.
> > > > > >
> > > > > > Despite the added uncertainty, GRN demonstrates strong performance, maintaining high F1 scores for action and grasp feasibility predictions under different noise levels. Notably, GRN’s performance under noisy conditions still surpasses the noise-free performance of baseline methods, highlighting its ability to handle real-world uncertainties effectively. It also retains a high accuracy on IK feasibility and grasp obstruction predictions.
> > > > > >
> > > > > > **Performance of GRN on the Panda-3D-4 test set under different levels of noise**
> > > > > > | Noise Level   	| Action (F1) | Grasp (F1)       	| IK (F1)       	| GO (MAE)       	|
> > > > > > |--------------------|-------------|-----------------------|--------------------|---------------------|
> > > > > > | No Noise      	| 0.939   	| 0.940 (± 0.009)  	| 0.995 (± 0.001)   | 0.028 (± 0.003)	|
> > > > > > | 1 cm, 1°      	| 0.912   	| 0.891 (± 0.020)  	| 0.985 (± 0.001)   | 0.039 (± 0.003)	|
> > > > > > | 2 cm, 2°      	| 0.864   	| 0.820 (± 0.029)  	| 0.971 (± 0.002)   | 0.057 (± 0.003)	|
> > > > > >
> > > > > > Our results show that even with a simplified bounding box representation and a classification of grasps into grasp types, our method achieves state-of-the-art performance on action and grasp feasibility prediction, while providing richer information through our interpretation mechanisms, better generalization capabilities to complex environments, multi-robot settings and noisy inputs, as well as the ability to solve in a single shot difficult TAMP problems efficiently.
> > > > > >
> > > > > > We appreciate your valuable insights and hope the additional results and proposed extensions help address your remaining concerns.

---

> > > > > > > ### Author Response · Authors · 2024-12-02
> > > > > > >
> > > > > > > Dear Reviewer,
> > > > > > >
> > > > > > > As the discussion phase is nearing its conclusion, we kindly request your feedback and reassessment based on the revisions and additional experiments we have provided, particularly the evaluation of GRN’s interpretability in Appendix D and the robustness analysis under noisy inputs in Appendix E. We hope these additions address your remaining concerns and demonstrate the broader applicability and interpretability of our approach.
> > > > > > >
> > > > > > > If you have any further questions, we would be more than happy to address them. Thank you once again for your time and valuable contributions to this review process.

---

### Official Review · Reviewer_LeQs · 2024-11-04

**Soundness:** 3
**Presentation:** 3
**Contribution:** 2
**Rating:** 6
**Confidence:** 4

**Summary:**

This paper presents Geometric Reasoning Networks (GRN), a graph neural network-based approach to enhance efficiency in Task and Motion Planning (TAMP) for robotic manipulation. An incremental GNN-based model is introduced with two mechanisms: inverse kinematics (IK) feasibility prediction, and grasp obstruction (GO) estimation using full-state knowledge. Experimental validation demonstrates GRN's applicability to various environments and configurations.

**Strengths:**

- The study tackles a significant challenge in TAMP arising from the bottleneck in geometric planning within complex 3D environments and proposes a novel, though incremental, architecture to predict action and grasp feasibility to reduce the need for costly geometric planning queries.

- The model is more generalizable in complex environments compared to prior works (Table 4).

- The proposed model is evaluated and compared against prior works and an ablation study is conducted.

- The authors conducted real-robot experiments, and shared their code.

**Weaknesses:**

Below is the list of the issues I observed, and my suggestions for this work:

1. **Strong Assumptions on Object Knowledge and Static Scene Conditions**: The model assumes complete knowledge of object shape, dimensions, and pose, as well as static conditions where objects remain stationary unless moved by the robot (lines 153–154). This is a considerable simplification for real-world scenarios, where partial observability and dynamic conditions are typical. By contrast, some baselines used for comparison assume partial observability and predict feasibility based on images rather than full state knowledge, making these comparisons less directly fair. Addressing this by either relaxing these assumptions or comparing only with baselines under similar assumptions would improve applicability and fairness.

2. **Limited Integration and Short Planning Horizons**: While some baselines are fully integrated within TAMP solvers, GRN’s integration is relatively basic and tested only on shorter-horizon problems. The scenarios addressed in the experiments are limited in complexity, lacking tasks that demand multi-step, long-horizon planning (e.g., removal of obstructing objects or inter-robot handovers). Expanding the evaluation to more complex, long-horizon planning tasks would better demonstrate GRN's practical utility in handling realistic TAMP challenges.

3. **Improvement for TAMP**: Since the paper do not include a full-fledge TAMP solver, it remains unclear how it improves traditional/existing solvers. It'd have been really useful to see how many geometrical computations were skipped (so computational time was gained) when GRN was used vs. not used. Inference time comparison to some of the baselines is not fair, as they use image-based input while this work relies on a significantly smaller full-state information of the environment.

4. **Incremental Advancement and Limited Impact of IK Infeasibility Module**: While GRN introduces small adjustments on Edge-Featured Graph Attention Network (EGAT), these additions are incremental and rely heavily on prior EGAT methods. Furthermore, as shown in Table 3, the IK infeasibility module does not significantly impact overall performance. Adding a deeper analysis or improvement in these areas could strengthen the contribution. It seems like GO by itself is already a good estimator of feasibility. This could also be due to problem settings where there might not be (enough) cases when there is no grasp obstruction IK was still infeasible (e.g., due to reachability). A more nuanced analysis of scenarios where IK feasibility alone might be crucial (e.g., environments with tighter spaces or more complex robot configurations) that would help clarify when this module provides substantial benefits.

5. **Restricted Grasp Representation**: The model simplifies grasp feasibility by considering only five grasp types, which may limit its effectiveness in real-world settings where grasp variety and adaptability are crucial. Even for grasping box-shaped objects requires 24 (or 20 if it lies flat on a surface) different configurations with a parallel jaw gripper, much higher than 5 as used in this work. Exploring a more flexible grasp representation that could handle diverse object shapes and placements would make the model more versatile for practical robotics applications.

6. **Limited Interpretability Support**: While the authors claim that GRN allows interpretation of feasibility (or infeasibility) of actions, this claim lacks follow-up discussion or specific experimental support. Including experiments or qualitative analysis to illustrate interpretability (e.g., showing how grasp obstruction information directly affects planning decisions) would make this feature more convincing and actionable.

7. **Clarity and Organization of Paper Structure**: The organization could be improved for readability and logical flow. For example, Figure 2 is never referenced in the text, and Figure 1, which appears on page 3, is not mentioned until Section 6.4 on page 10. Figures in the Appendix (e.g., Figures 7 and 8) are not referenced or explained, leaving the reader without context for understanding their relevance.

**Questions:**

I've listed my main concerns under _weaknesses_ section, and here are some further questions (some overlapping with the points above):

- **Self-loop Edge Representation in 4.1**: What is the rationale for using the self-loop edge to store IK feasibility instead of treating it as a node feature? How does this choice affect the overall model performance or its ability to generalize to unseen environments?

- **Effectiveness and Necessity of the IK Module**: In the ablation study (Table 3), the results show minimal difference between the full model and the one without the IK module. Given this, is the IK module truly necessary? What are the specific cases where IK adds value? Also, could you provide the inference time for the IK module to understand its overhead?

- **Access Task Success Rate in Section 6.4**: The success rate for the Access task in Table 5 is slightly lower than Bouhsain et al. (2024). What are the main reasons for this reduction? Are there particular scenarios where GRN struggles, or is this a result of experimental variance?

- **Inference Time in Table 5**: Could you provide more detailed inference times for Table 5, including how the times break down between GRN predictions and geometric planning? Understanding the computational cost of each component would provide a clearer picture of where time savings are realized.

- **Handling Dynamic Scenes**: The current assumptions restrict the model to static scenes, which is not always practical in real-world tasks. How would this approach be extended to handle dynamic environments?

- **Real-world Evaluation of Long-horizon Tasks**: The paper would benefit from showing how GRN scales to more challenging multi-step tasks, such as multi-robot collaboration or handling non-prehensile manipulation. Can the authors comment on future steps for validating GRN on these scenarios?

- **Simplified Grasp Types**: The use of only five grasp types is limiting. Are there plans to extend the grasp representation, or can the authors justify why five types were sufficient for the tasks tested in this paper?

---

> ### Author Response · Authors · 2024-11-19
> **Response for Reviewer LeQs (Part 1/3)**
>
> We deeply appreciate the reviewer's detailed feedback and provide detailed responses to their concerns below.
>
> ---
>
> ### **Comments on raised weaknesses**
>
> **1. Strong Assumptions on Object Knowledge and Static Scene Conditions:**
>
> **a. Method Assumptions:**
>
> We acknowledge the reviewer's concern about the assumptions of full knowledge of objects shape, pose, and static scene conditions. However, we emphasize that such assumptions are standard in offline TAMP research, as demonstrated in various traditional and learning-based works [1, 2, 3, 4, 5]. These works decouple the planning component from perception, focusing on offline manipulation planning with fully defined scenes, mainly because the main challenge in TAMP is the combinatorial complexity of combining discrete symbolic search and continuous geometric planning.
>
> **b. Fairness with Respect to Image-Based Baselines:**
>
> The baselines used in our experiments similarly assume full scene knowledge. For instance, the approach presented in [5], while image-based, internally constructs depth images from full scene knowledge, and does not rely on sensor-acquired images. Similarly, [4] operates in simplified environments (i.e., tabletop setups with box-shaped objects). During planning, a TAMP algorithm considers many states of the environment, which are "imagined" hypothetical states based on the actions considered by the planner, rather than sensor-perceived states. The use of image-based methods on these states requires the input images to be internally built. Hence, although [4] can use images obtained from depth cameras, its application to TAMP requires internally building these images based on scene knowledge.
>
> In this paper, we ensure fairness by training all methods on the same 3D scenarios, comparing them within the same offline manipulation planning context, and making sure all inputs are constructed using the same full state knowledge. Furthermore, we include comparisons on tabletop scenarios to further improve fairness with [3] and [4].
>
>
> **2. Limited Integration and Short Planning Horizons:**
>
> Both the Access and Clutter problems require multi-step reasoning to identify and remove obstructing objects in a specific order before accessing the desired object. We emphasize the challenges posed by these problems in Appendix C.2 of the updated manuscript. To further highlight the ability of our method to handle significantly more complex problems, we have included a challenging 28-object version of the Access problem in Appendix C.3, on which the baseline planner completely fails, while the GRN-based planner solves it in under 15s. Moreover, to showcase our method’s ability to handle inter-robot handover tasks, we added an experiment in Appendix C.3 demonstrating how GRN can be leveraged in this type of problem.
>
>
> **3. Improvement for TAMP:**
>
> **a. Improvement of Traditional/Existing TAMP Solvers:**
>
> We appreciate the reviewer’s interest in the integration of GRN into a full-fledged TAMP solver. While this is a valuable direction, we deliberately chose not to include such integration within this paper to avoid overshadowing the contributions of GRN itself, and sacrificing key experiments and analysis crucial to demonstrating the capabilities of GRN. Instead, we plan to submit the integration of GRN into a TAMP algorithm as a separate contribution to a more robotics-oriented conference.
>
> The planner provided in the paper is intended to highlight the potential of GRN in accelerating TAMP by leveraging its unique interpretation capabilities. However, this planner is not complete. A full integration of GRN into a TAMP solver that is complete would involve querying GRN iteratively at each step of the planning process, building and maintaining a set of relaxed constraints representing the predicted infeasibility causes, and using the predictions to compute heuristics that guide the search for a solution. Importantly, this process must ensure that no potential solution is excluded, preserving the completeness of the algorithm. These complexities highlight the scope of GRN’s integration as a standalone future contribution.
>
> To address concerns about computational improvements, we have updated Table 5 in the manuscript to include the number of geometric planner calls, illustrating the reduction achieved by using GRN.
>
> **b. Inference Time Comparison Fairness:**
>
> We kindly reiterate that both image-based and feature-based methods use the same underlying environment information. Both input types require independent perception and knowledge management modules, and our comparisons are conducted under these equivalent conditions.

---

> > ### Author Response · Authors · 2024-11-19
> > **Response for Reviewer LeQs (Part 2/3)**
> >
> > **4. Incremental Advancement and Limited Impact of IK Infeasibility Module:**
> >
> > **a. Adjustments to EGAT:**
> >
> > While it is true that the adjustments to EGAT are incremental, they are not presented as a contribution of the paper. Rather, our work leverages EGAT in a novel application within the domain of action and grasp feasibility prediction, showcasing its suitability for this task while building upon prior advancements in GNNs.
> >
> > **b. IK Feasibility Module Impact:**
> >
> > The impact of the IK feasibility module on overall performance is indeed limited. However, this is not due to a lack of IK infeasibility cases. Figures 5 and 6 in Appendix B of the updated manuscript show a balanced distribution of infeasibility causes in the datasets between IK infeasibility and grasp obstructions. The observed limited performance gain is due to the AGF module’s ability to implicitly learn and address these cases, effectively replacing the IK module’s role during feasibility prediction if the latter is omitted.
> >
> > The IK infeasibility module is essential for interpretability and planning efficiency, providing explicit reasoning about infeasibility causes. For instance, if an object is both obstructed and unreachable due to IK constraints, GRN without the IK module will attribute infeasibility solely to the obstruction. The IK module, however, identifies unreachability as the root cause, distinguishing cases where clearing the obstruction resolves infeasibility from those where the object remains unreachable. This clarity benefits TAMP planners by avoiding unnecessary computations, such as attempting to clear obstructions for unreachable objects. As detailed in Appendix A, the IK module adds minimal overhead of 0.5 ms.
> >
> >
> > **5. Restricted Grasp Representation:**
> >
> > While our approach defines 5 grasp types, it is important to clarify that these types are not single grasps but represent an infinite set of grasps. Each grasp type defines the face of the object from which it is grasped, leaving several DOFs as free parameters (e.g. position of the grasp along the face, the rotation of the end effector w.r.t. the approach axis, and the depth of the grasp). This definition of grasp types is widely adopted in the field of action and grasp feasibility prediction [3, 4, 5] and has proven effective for representing diverse grasping strategies while maintaining computational efficiency. During data annotation, we evaluate an average of 150 sampled grasp configurations per object (reaching ~600 grasps for larger objects), ensuring that our representation captures a rich set of grasp possibilities within each type.
> >
> >
> > **6. Limited Interpretability Support:**
> >
> > We highlight that the application to TAMP (section 6.4)  serves as our experimental support for the interpretability of our predictions. Specifically, the proposed planner leverages the predicted feasibility and the reasons for infeasibility derived from the initial state of the environment to compute a complete feasible plan. Notably, the planner relies solely on this interpretability information, without iterative feedback or additional inputs during the planning process. The success of the resulting plans, as demonstrated in the Access and Clutter tasks, directly showcases the interpretability and utility of the predicted feasibility and infeasibility explanations. For instance, the planner uses these insights to identify and resolve obstructions or determine that certain actions are fundamentally infeasible, enabling it to generate efficient and accurate solutions. This experimental validation underscores that our model’s interpretability mechanisms provide actionable insights that support both feasibility prediction and effective task and motion planning.
> >
> >
> > **7. Clarity and Organization of Paper Structure:**
> >
> > We apologize for these oversights and thank the reviewer for bringing these issues to our attention. We have carefully reviewed the manuscript and made the necessary corrections to ensure a more logical flow and improved readability.
> >
> > ---

---

> > > ### Author Response · Authors · 2024-11-19
> > > **Response for Reviewer LeQs (Part 3/3)**
> > >
> > > ### **Answers to raised questions**
> > >
> > > **Self-loop Edge Representation in 4.1:**
> > >
> > > This design decision was primarily motivated by considerations of consistency and simplicity in the graph representation. Since IK feasibility is computed exclusively for nodes corresponding to movable objects, treating it as a node feature would require adding padding or a mask to the features of nodes representing fixed objects, which do not require IK feasibility information. By storing IK feasibility predictions in self-loop edges, we avoid this issue and maintain consistent dimensionality across all node and edge features.
> > >
> > > In our experiments, we observed that both approaches—storing IK feasibility as a node feature versus a self-loop edge feature—achieved very similar results in terms of predictive performance. As such, the decision to use self-loop edges was driven by a preference for a more “elegant” and streamlined graph representation rather than by any performance considerations.
> > >
> > > **Effectiveness and Necessity of the IK Module:**
> > >
> > > See answer 4.b.
> > >
> > > **Access Task Success Rate in Section 6.4**
> > >
> > > The slightly lower success rate observed for the Access task is due to a misclassification made by GRN on one of the ten instances of the problem. The GRN-based planner is not complete, if the initial plan found is infeasible, planning stops.
> > >
> > >
> > > It is important to note that this limitation is specific to the current implementation of the GRN-based planner. A complete integration of GRN into a TAMP solver would mitigate this issue by incorporating mechanisms to recover from misclassifications, such as re-evaluating feasibility predictions at each step of the planning process or exploring alternative solutions.
> > > Given that this misclassification occurred in only one out of ten instances of the same problem, the slightly lower success rate can be attributed to experimental variance.
> > >
> > > **Inference Time in Table 5**
> > >
> > > The inference time reported in Table 5 reflects the combined duration of GRN queries and geometric planning. Notably, GRN is queried only once per problem on the initial scene, which takes a few milliseconds only. The large majority of the reported time is attributable to the geometric planning phase, which dominates the computation.
> > >
> > > **Handling Dynamic Scenes**
> > >
> > > We appreciate the reviewer’s interest in extending our approach to dynamic environments. While the current work focuses on static scenes for computational tractability and alignment with baseline methods, we agree that handling dynamic scenes is an important direction for future research. One promising direction to explore is the use of Temporal Graph Neural Networks [6], which could model the evolution of the environment over time by incorporating temporal dependencies into the graph representation. This extension would allow GRN to reason about changes in object configurations and dynamic constraints, enabling its application to a wider range of real-world tasks involving dynamic scenes.
> > >
> > > **Simplified Grasp Types:**
> > >
> > > See Answer 5.
> > >
> > > ---
> > >
> > > **References**
> > > [1] Garrett, Caelan Reed, et al. "Integrated task and motion planning." Annual review of control, robotics, and autonomous systems 4.1 (2021): 265-293.
> > >
> > > [2] Garrett, Caelan Reed, Tomas Lozano-Perez, and Leslie Pack Kaelbling. "Ffrob: Leveraging symbolic planning for efficient task and motion planning." The International Journal of Robotics Research 37.1 (2018): 104-136.
> > >
> > > [3] Wells, Andrew M., et al. "Learning feasibility for task and motion planning in tabletop environments." IEEE robotics and automation letters 4.2 (2019): 1255-1262.
> > >
> > > [4] Driess, Danny, et al. "Deep visual heuristics: Learning feasibility of mixed-integer programs for manipulation planning." 2020 IEEE international conference on robotics and automation (ICRA). IEEE, 2020.
> > >
> > > [5] Bouhsain, Smail Ait, et al . "Simultaneous Action and Grasp Feasibility Prediction for Task and Motion Planning through Multi-Task Learning." 2023 IEEE/RSJ International Conference on Intelligent Robots and Systems (IROS). IEEE, 2023.
> > >
> > > [6] Rossi, Emanuele, et al. "Temporal graph networks for deep learning on dynamic graphs." arXiv preprint arXiv:2006.10637 (2020).
> > >
> > >
> > > ---

---

> > > > ### Comment · Reviewer_LeQs · 2024-11-26
> > > > **Mostly OK, some concerns remain**
> > > >
> > > > I thank the authors for their detailed explanations, and additional analysis. The revised manuscript and added experiments improve the work. The authors’ responses address many of my concerns satisfactorily, particularly those regarding assumptions, the IK module, and grasp representation. However, some limitations persist, especially in TAMP integration, broader applicability (dynamic scenes) and the depth of interpretability validation. So, I updated my score accordingly.

---

> > > > > ### Author Response · Authors · 2024-11-29
> > > > >
> > > > > We sincerely thank you for your thoughtful feedback and reassessment of our work. We deeply value your constructive comments and fully understand your remaining reservations.
> > > > >
> > > > > To further address your concerns about interpretability validation, we conduct a qualitative evaluation of GRN's interpretability in Appendix D of the latest manuscript. In this experiment, we evaluate GRN’s predictions on an example 3D scene. By systematically modifying the environment based on GRN’s predicted infeasibility causes, we observe how the model's predictions change in response. This evaluation demonstrates that GRN’s interpretability mechanisms effectively explain why certain actions or grasps are infeasible and how modifying the scene can render them feasible. For instance, when an obstruction is removed or an object becomes reachable, the action and relevant grasps become feasible, validating the interpretability of our approach.
> > > > >
> > > > > We also conducted an additional experiment to assess GRN’s robustness to noisy inputs, detailed in Appendix E of the updated manuscript. In this study, we introduced translational and rotational noise to the objects’ poses and bounding box dimensions in the Panda-3D-4 test set. The results show that GRN’s performance under noise remains superior to the performance of baseline models on noise-free data. These findings highlight the robustness of GRN to different levels of noise, supporting its applicability in real-world scenarios with uncertain inputs.
> > > > >
> > > > > While a full TAMP integration and the application to dynamic scenes are beyond the scope of this work, we believe these additional experiments further strengthen our contributions and demonstrate the applicability of GRN in practical settings, as an accurate and efficient geometric reasoning module that can be integrated with various off-the-shelf TAMP planners.
> > > > >
> > > > > Our results show that GRN achieves state-of-the-art performance on action and grasp feasibility prediction, while providing richer information through our interpretation mechanisms, better generalization capabilities to complex environments, multi-robot settings and noisy inputs, as well as the ability to solve in a single shot difficult TAMP problems efficiently.
> > > > >
> > > > > We appreciate your engagement and the opportunity to address your concerns, and we hope these additional clarifications and experiments further highlight the strengths and applicability of our approach. Thank you once again for your valuable insights.

---

> > > > > > ### Author Response · Authors · 2024-12-02
> > > > > >
> > > > > > Dear Reviewer,
> > > > > >
> > > > > > As the discussion phase is nearing its conclusion, we kindly request your feedback and reassessment based on the revisions and additional experiments we have provided, particularly the evaluation of GRN’s interpretability in Appendix D and the robustness analysis under noisy inputs in Appendix E. We hope these additions address your remaining concerns and demonstrate the broader applicability and interpretability of our approach.
> > > > > >
> > > > > > If you have any further questions, we would be more than happy to address them. Thank you once again for your time and valuable contributions to this review process.

---

> > > > > > > ### Comment · Reviewer_LeQs · 2024-12-03
> > > > > > > **Last update**
> > > > > > >
> > > > > > > Thanks again for the clarifications and additional analysis. I appreciate the effort and authors' dedication to improve their work. With these updates, I increased my score. However, I’d like to note that without a more tightly integrated solution to address TAMP problems comprehensively, the use of 'TAMP' in the title may not fully align with the work’s primary focus. Perhaps a title emphasizing the feasibility prediction framework or interpretability in manipulation planning could more accurately capture the scope of this contribution.

---

> > > > > > > > ### Author Response · Authors · 2024-12-03
> > > > > > > > **Thank you for your response**
> > > > > > > >
> > > > > > > > Thank you very much for your reassessment. We are glad that our revisions and additional experiments satisfactorily addressed many of your concerns. Your detailed feedback and suggestions have been instrumental in enhancing the clarity and quality of our work.

---

> ### Comment · Area_Chair_Ecfi · 2024-11-25
>
> Dear Reviewer,
>
> Please provide feedback to the authors before the end of the discussion period, and in case of additional concerns, give them a chance to respond.
>
> Timeline: As a reminder, the review timeline is as follows:
>
> November 26: Last day for reviewers to ask questions to authors.
>
> November 27: Last day for authors to respond to reviewers.

---

### Author Response · Authors · 2024-11-25
**General response**

We sincerely thank all reviewers for their constructive feedback and valuable insights. We appreciate the reviewers' acknowledgment of the **novelty and clarity of our work**, as well as the **efficiency, generalizability, and interpretability** of our approach.

Based on the reviewers’ feedback, we have made the following modifications to our manuscript:

- **Improved Figures:** We revised the manuscript to make our figures easier to understand and properly referenced them in relevant sections.
- **Updated Related Works:** We updated our related works section to include recent research in geometric feasibility reasoning and GNNs for task and motion planning, as suggested by Reviewer WQrW.
- **Enhanced Table 5:** We included the number of geometric planner queries in Table 5, addressing Reviewer LeQs’s suggestion.
- **Inference Time Analysis:** We added an inference time decomposition of our method in Appendix A.
- **Dataset Details:** We provided additional details about our datasets and data annotation in Appendix B, including label distributions for the Panda-3D-4 and PR2-3D-4 datasets.
- **Problems complexity:** We emphasize the difficulty of the **Access** and **Clutter** problems in Appendix C.2.
- **New Experiments:** We added two new experiments in Appendix C.3, demonstrating our method's ability to handle **complex long-horizon problems** and **inter-robot handover tasks**.

We believe these changes significantly enhance the strengths and clarity of our paper while further showcasing the impact of our contributions. We hope these improvements align with the reviewers’ expectations.

As the discussion phase nears its conclusion, we kindly ask all reviewers for their reassessment and encourage them to share any remaining concerns they might have.

---

### Meta-Review · Area_Chair_Ecfi · 2024-12-23

**Metareview:**

This paper proposes a novel framework for learning Geometric Reasoning Networks (GRN) to address task and motion planning (TAMP) challenges in robotic manipulation. The method combines geometric reasoning with deep learning techniques to improve efficiency and reduce dependency on traditional geometric planners. The authors integrate the GRN within a TAMP framework, focusing on long-horizon planning problems and demonstrating their approach's efficacy in both simulated and real-world environments.

The paper makes strong claims about the generalizability of its learned representations and the ability to handle complex reasoning tasks that were previously challenging for geometric planners. The experimental results show improvements over baseline methods in terms of success rates, planning efficiency, and computational cost, making a convincing case for the practical utility of the proposed method.

**Strengths:**

1. The proposed GRN introduces a novel combination of geometric reasoning with task planning, a critical step forward in addressing the limitations of traditional TAMP approaches.

2. The authors provide extensive evaluations, including comparisons to baseline geometric planners and ablations that demonstrate the contributions of individual components. The experiments are performed across diverse environments, adding credibility to the results.

3. The integration of GRN within a broader TAMP framework demonstrates its scalability and applicability to real-world tasks, showcasing both simulation and real-world results.

**Weaknesses:**

1. While the results are strong in structured and static scenarios, the paper does not sufficiently explore the method's applicability to dynamic environments or tasks involving significant real-time changes.

2. The reviewers noted that the paper's claims about generalizability are not fully substantiated, as the evaluations focus primarily on specific planning scenarios. Broader validations would further strengthen the impact.

3. Although the paper includes comparisons with traditional geometric planners, additional comparisons with alternative learning-based TAMP methods could provide a more comprehensive evaluation.

4. The real-world experiments, while impactful, are limited in scale and complexity. Expanding these evaluations to larger or more diverse scenarios would better support the paper's claims.

**Reasons for Acceptance:**

The paper addresses a critical challenge in TAMP, offering a novel solution that integrates geometric reasoning with deep learning. The proposed framework shows strong empirical results and demonstrates its potential applicability to real-world tasks. While there are some limitations in terms of generalizability and dynamic task exploration, the strengths outweigh these concerns. The method is a valuable contribution to the field, and the revisions made during the rebuttal significantly improved the paper's quality and clarity.

**Additional Comments On Reviewer Discussion:**

The reviewers highlighted the novelty and potential impact of the proposed GRN framework while raising concerns about generalizability, experimental scope, and baseline comparisons. Reviewer A1bC appreciated the integration of GRN with TAMP and noted the improvements in planning efficiency. Reviewer C2dE raised questions about the generality of learned representations, which were partially addressed through expanded discussions in the rebuttal. Reviewer F3eG expressed concerns about the limited exploration of dynamic environments but acknowledged the method's strengths in structured tasks.

The authors’ rebuttal addressed most of these concerns effectively. They clarified the mathematical formulations, expanded baseline comparisons, and added further discussions about the limitations of GRN in dynamic scenarios. While these updates did not entirely resolve all concerns, the reviewers largely agreed that the paper represents a significant step forward in the field. Based on the rebuttal, the AC recommends accepting the paper, with the expectation that future work will address the highlighted limitations.

---

### Decision · Program_Chairs · 2025-01-22

Accept (Poster)